# DfPO: Degeneration-free Policy Optimization via Action Masking in Natural Language Action Spaces

## Abstract

As the pre-training objectives (e.g., next token prediction) of language models (LMs) are inherently not aligned with task scores, optimizing LMs to achieve higher downstream task scores is essential. One of the promising approaches is to fine-tune LMs by using reinforcement learning (RL). However, conventional RL methods based on PPO and a penalty of KL divergence are vulnerable to the text degeneration problem which LMs do not generate natural texts anymore after RL fine-tuning. To address this problem, we provide **Degeneration-free Policy Optimization (DfPO)** that can fine-tune LMs to generate texts that achieve improved downstream task scores, while preserving the naturalness of the generated texts. To achieve this, we introduce *action-masked policy* with which a behavior policy can avoid to select tokens that potentially make policy optimization unexpected. Then, we devise *clipped advantage functions* to separately perform likelihood maximization and minimization, conditioned on texts sampled from the action-masked policy. Our experiments on the GRUE benchmark demonstrate that DfPO successfully improves the downstream task scores, while preserving the naturalness of the generated texts. Moreover, even DfPO does not perform hyperparameter search, it achieves similar performance to PPO and NLPO which require additional hyperparameter search for the penalty ratio of KL divergence.

## 1 Introduction

Although pre-trained language models (LMs) have achieved remarkable success in various NLP tasks (Ouyang et al., 2022; Glaese et al., 2022; Bai et al., 2022; Stiennon et al., 2020; Nakano et al., 2022), fine-tuning on downstream tasks is essential to achieve higher task scores. Since the pre-training and supervised fine-tuning objectives (e.g., next token prediction (Radford et al., 2019)) of LMs are inherently not maximizing the task scores, LMs fail to learn an optimal behavior. One of the promising approaches to fine-tune the LMs is reinforcement learning (RL) (Christiano et al., 2017). Recently, reinforcement learning from human feedback (RLHF) (Christiano et al., 2017; Liang et al., 2022; Kim et al., 2023; Ouyang et al., 2022) methods, which learn a reward model from human feedback and then fine-tune LMs through reinforcement learning, has successfully achieved to fine-tune the LMs using RL (Ouyang et al., 2022; Glaese et al., 2022; Bai et al., 2022; Stiennon et al., 2020; Nakano et al., 2022). However, optimizing LMs against a given reward model by using RL is yet challenging due to the *degeneration problem* which generates responses diverging from human language.

When optimizing LMs with RL, most of the existing methods mainly use PPO (Schulman et al., 2017) for optimizing an LM policy, and a penalty of KL divergence between the reference LM and an optimized LM for preserving the naturalness of generated texts (Ouyang et al., 2022; Ramamurthy et al., 2023). However, conventional RL methods based on PPO and a KL divergence penalty are vulnerable to the text degeneration problem that LMs do not generate natural texts anymore after RL fine-tuning. We illustrate the text degeneration problem in Figure 1. We carefully conjecture that a penalty of KL divergence often does not work with respect to the penalty ratio $\beta$, since PPO is a simplified algorithm from TRPO Schulman et al. (2015) that guarantees to maximize target task rewards. In other words, it is important to balance two different objectives: (1) maximizing task rewards and (2) minimizing the KL divergence, while preventing one objective to dominate the

overall quantity. Especially in language generation tasks, confining the probability distribution of LMs within a certain level is more important than simply maximizing the task rewards.

To address this problem, in this paper, we provide **Degeneration-free Policy Optimization (DfPO)** that can fine-tune LMs to generate texts that effectively achieve higher downstream task scores, while preserving the naturalness of the generated texts. To achieve this, we introduce *action-masked policy* with which a behavior policy can avoid to select tokens that potentially make policy optimization unexpected. Then, we reformulate policy optimization as stable likelihood max/minimization with *clipped advantage functions* that eventually mitigates excessive task reward optimization. More importantly, DfPO does not require any sensitive hyperparameters such as the penalty ratio of KL divergence used in conventional RL methods. Our experiments on the GRUE Ramamurthy et al. (2023) benchmark demonstrate that DfPO is much more effective in optimizing task scores than PPO and NLPO Ramamurthy et al. (2023), while preserving the naturalness of generated texts.

The contributions of this paper can be summarized as follows:

- We find that conventional RL fine-tuning methods that use PPO Schulman et al. (2017) and a penalty of KL divergence are vulnerable to result in the text degeneration problem that LMs do not generate natural texts anymore after RL fine-tuning (see Figure 1).

- We introduce Degeneration-free Policy Optimization (DfPO) that can fine-tune LMs to generate texts that achieve higher downstream task scores, while preserving the naturalness of the generated texts. To achieve this, we develop *action-masked policy* (see Table 1 and Eq. 5, 6) and stable likelihood max/minimization with *clipped advantage functions* (see Eq. 7).

- We demonstrate that, even DfPO does not perform hyperparameter search, it achieves similar performance to PPO Schulman et al. (2017) and NLPO Ramamurthy et al. (2023) which require additional hyperparameter search for the penalty ratio of KL divergence on the GRUE Ramamurthy et al. (2023) benchmark (see Figure 3 and Table 2).

- We show that DfPO can improve task scores while preserving the naturalness of the generated texts, even when using a large language model (GPT-J-6B) as a policy (see Figure 6).

## 2 PRELIMINARIES

### 2.1 REINFORCEMENT LEARNING FOR LANGUAGE MODELS

We consider the generative NLP tasks that can be modeled as a Markov decision process (MDP) defined by tuple $M = \langle S, A, T, r, p_0, \gamma \rangle$ (Sutton & Barto, 1998), where $S$ is the set of states $s$ (a sequence of word tokens), $A$ is the set of actions $a$ (a next word token), $T : S \times A \to \Delta(S)$ is the transition probability, $r : S \times A \to \mathbb{R}$ is the reward function, $p_0 \in \Delta(S)$ is the distribution of the initial state, and $\gamma \in [0, 1)$ is the discount factor. The policy $\pi(a|s)$ is a mapping from state to a probability distribution over $A$, which can be naturally modeled as language models. The value function, action-value function, and advantage function are defined as $V^\pi(s) := \mathbb{E}_{s_0, a_0, \cdots \sim \pi}[\sum_t^\infty \gamma^t r(s_t, a_t)|s_0 = s]$, $Q^\pi(s, a) := r(s, a) + \gamma \mathbb{E}_{s' \sim T(s,a)}[V^\pi(s')]$, and $A^\pi(s, a) := Q^\pi(s, a) - V^\pi(s)$, respectively. Unlike standard RL problems that aim to maximize only cumulative rewards, in NLP tasks, the goal is to find an optimal policy that maximizes cumulative rewards while preserving the language quality (i.e. naturalness) of generated texts.

### 2.2 POLICY GRADIENT WITH KL-REGULARIZATION PENALTY

Policy gradient algorithms are widely employed in reinforcement learning to maximize the expected cumulative rewards $\mathbb{E}_{s_0, a_0, \cdots \sim \pi}[\sum_{t=0}^\infty \gamma^t r(s_t, a_t)]$. In this paper, we use the following formulation based on the advantage function, which represents the most commonly used form of the policy gradient estimator:

$$\nabla_\theta J(\theta) = \mathbb{E}_{(s,a) \sim d^{\pi_\theta}} \left[ A^{\pi_\theta}(s, a) \nabla_\theta \log \pi_\theta(a|s) \right], \tag{1}$$

where $d^\pi(s) := (1 - \gamma) \sum_{t=0}^\infty \gamma^t \Pr(s_t = s)$ is a stationary distribution with $s_0 \sim p_0$ and the actions are chosen according to $\pi$, and $d^\pi(s, a) := d^\pi(s)\pi(a|s)$. However, conventional RL algorithms that focus solely on maximizing single-task rewards easily result in *reward hacking* behaviors, which correspond to the *degeneration problems* in generative NLP tasks. The current best-performing RL

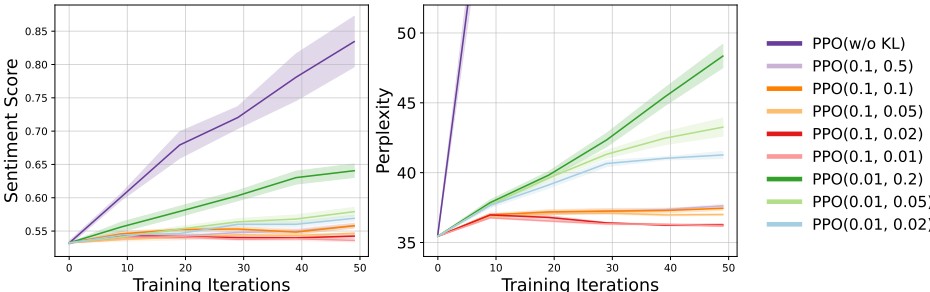

Figure 1: Averaged learning curve over 5 runs of PPO with KL-regularization penalty on IMDB text continuation task for varying KL coefficient and target KL. PPO ($\beta$, $\mathrm{KL_{target}}$) indicates the PPO that considers the KL-regularization as a reward penalty with KL coefficient $\beta$ and target KL. The goal of the IMDB text continuation task is to learn a policy that maximizes the sentiment score (i.e. task reward) while preserving the perplexity score (i.e. naturalness). However, as shown in the results, PPO with KL-regularization penalty shows very sensitive performance on both sentiment score and perplexity score to hyperparameter. We also provide the results of NLPO with KL-regularization penalty in Appendix B.

algorithm for generative NLP tasks utilize a KL-divergence penalty between the current policy $\pi_\theta$ and the reference policy $\pi_0$ as follows:

$$\underset{\theta}{\text{maximize}} \quad \mathbb{E}_{(s,a)\sim d^{\pi_\theta}}\left[A^{\pi_\theta}(s,a)\right] \tag{2}$$

$$\text{subject to} \quad \mathbb{E}_{s\sim d^{\pi_\theta}}\left[\mathrm{KL}(\pi_\theta(\cdot|s)\|\pi_0(\cdot|s))\right] \leq \delta, \tag{3}$$

where $\delta > 0$ is a hyperparameter. Based on the Lagrangian, we obtain the following unconstrained optimization for the constrained optimization problem in (2-3):

$$\underset{\theta}{\text{maximize}}\ \mathbb{E}_{(s,a)\sim d^{\pi_\theta}}\left[A^{\pi_\theta}(s,a) - \beta\mathrm{KL}(\pi_\theta(\cdot|s)\|\pi_0(\cdot|s))\right] = \mathbb{E}_{d^{\pi_\theta}}\left[A^{\pi_\theta}(s,a) + \beta u_\theta(s,a)\right], \tag{4}$$

where $\beta \geq 0$ is a fixed hyperparameter rather than a Lagrangian multiplier as in previous studies, and $u_\theta(s,a) := \log\frac{\pi_0(a|s)}{\pi_\theta(a|s)}$. However, as discussed in prior works (Ramamurthy et al., 2023; Ouyang et al., 2022; Ziegler et al., 2019), optimizing the policy through the KL-penalized objective (4) is very sensitive to the hyperparameter $\beta$. Figure 1 shows the results of PPO with various $\beta$ on sentiment score (i.e. task performance) and perplexity (i.e. naturalness). This sensitivity to hyperparameters can be especially troublesome for large-scale language models that require *massive computational costs* and also cause *ambiguity in model selection*, as shown in Figure 1.

## 3 DEGENERATION-FREE POLICY OPTIMIZATION

In this section, we present **Degeneration-free Policy Optimization (DfPO)** that can fine-tune LMs to generate texts that achieve improved downstream task scores, while preserving the naturalness of the generated texts, without any additional hyperparameter search. First, we investigate why the degeneration problem occurs in policy gradient algorithms under the perspective of likelihood max/minimization. Then, we present our algorithm, DfPO, which mainly consists of 1) *action-masked policy* with which a behavior policy can avoid to select tokens that potentially make policy optimization unexpected, 2) *clipped advantage functions* to separately perform likelihood max/minimization conditioned on texts sampled from the action-masked policy.

### 3.1 UNDERSTANDING POLICY GRADIENT VIA LIKELIHOOD MAX/MINIMIZATION

Before presenting our algorithm, we first investigate why the degeneration problem occurs in policy gradient algorithms under the perspective of likelihood max/minimization. Intuitively, this policy gradient update can be interpreted as the likelihood max/minimization: gradient update through Eq. 1 increases the likelihood of actions that are better than the average behavior of the current policy (i.e. positive advantages) and decreases the likelihood of actions that are worse than the average behavior of the current policy (i.e. negative advantages). However, when considering regularized optimization as in Eq. 4, it is challenging to determine whether the actions are better or worse for the advantage $A^{\pi_\theta}$ and the log ratio $u_\theta$.

Table 1: Summary for all types of samples used in policy gradient update. The green area represents the state-action pairs that both advantage $A^{\pi_\theta}$ and log ratio $u_\theta$ are positive or negative, which are desirable samples for improving both task and naturalness scores. On the other hand, the red area represents the state-action pairs that have opposite directions for improving task score and naturalness score, which are undesirable samples that can cause degeneration problems in the policy update.

| | $A^{\pi_\theta}(s,a) > 0$ | $A^{\pi_\theta}(s,a) < 0$ |
|---|---|---|
| $u_\theta(s,a) > 0$ | Likelihood Maximization | $\|u_\theta(s,a)\| > \|A^{\pi_\theta}(s,a)\|$ or $\|u_\theta(s,a)\| < \|A^{\pi_\theta}(s,a)\|$ |
| $u_\theta(s,a) < 0$ | $\|u_\theta(s,a)\| > \|A^{\pi_\theta}(s,a)\|$ or $\|u_\theta(s,a)\| < \|A^{\pi_\theta}(s,a)\|$ | Likelihood Minimization |

Table 1 summarizes cases of all state-action samples used in the policy gradient update, according to the sign of the advantage and the log ratio. First, state-action pairs corresponding to the green area, where both $A^{\pi_\theta}$ and $u_\theta$ are positive or negative, are desirable samples that can update the policy to improve both task performance and naturalness through policy gradient. On the other hand, state-action pairs corresponding to the red area, which have opposite directions for improving task performance and naturalness, are undesirable samples that can cause degeneration problems in the policy gradient update. Depending on the magnitude of the absolute values of each advantage, a degeneration problem occurs or the task performance deteriorates. Therefore, if we perform a policy gradient update using only desirable samples in the green area, we can learn a policy that maximizes task reward while preserving the naturalness of the generated texts. However, if we simply generate samples using the current policy $\pi_\theta$ as the rollout policy, the proportion of samples in the green area is inevitably very low, which is inefficient in terms of sample efficiency for learning.

### 3.2 DEFINING ACTION-MASKED POLICY WITH REFERENCE POLICY

In order to efficiently generate only desirable samples corresponding to the green area in Table 1, we introduce action-masked policies, which are inspired by prior works for action masking to deal with combinatorial action space (Ammanabrolu & Hausknecht, 2020; Ramamurthy et al., 2023; Zahavy et al., 2018). Unlike prior methods that constrain the action spaces with top-$k$ or top-$p$ sampling, we adopt the relationship between the current policy and reference policy to constrain the action spaces of the behavior policy. More formally, we define *positive action-masked policy* $\pi^+_{\text{mask}}$ and *negative action-masked policy* $\pi^-_{\text{mask}}$ according to the relationship between the log-probabilities of the reference policy and current policy as follows:

$$\pi^+_{\text{mask}}(\cdot|s, \pi_\theta, \pi_0) := \begin{cases} \pi_\theta(\cdot|s)/Z^+ & \text{if } u_\theta = \log \pi_0 - \log \pi_\theta > 0 \\ 0 & \text{otherwise} \end{cases}, \quad (5)$$

$$\pi^-_{\text{mask}}(\cdot|s, \pi_\theta, \pi_0) := \begin{cases} \pi_\theta(\cdot|s)/Z^- & \text{if } u_\theta = \log \pi_0 - \log \pi_\theta < 0 \\ 0 & \text{otherwise} \end{cases}, \quad (6)$$

where $Z^+$ and $Z^-$ are normalizing constants and $\pi_0$ is a reference policy. Intuitively, the positive action-masked policy masks out the actions that can cause degeneration problems when their likelihood is increased, and the negative masked policy masks out the actions that can cause degeneration problems when their likelihood is decreased. In other words, the positive action-masked policy considers only those actions for which the $u_\theta$ is positive among the actions of the current policy, and the negative action-masked policy considers only those actions for which the $u_\theta$ is negative among the actions of the current policy.

### 3.3 LIKELIHOOD MAX/MINIMIZATION WITH CLIPPED ADVANTAGE FUNCTIONS

Using the action-masked policies defined above, we can separately obtain state-action pairs that need to increase and decrease the likelihood in terms of naturalness. Now, for each state-action pair generated from action-masked policies, it is necessary to determine whether the likelihood should be increased or decreased in order to improve the policy in terms of task performance. We

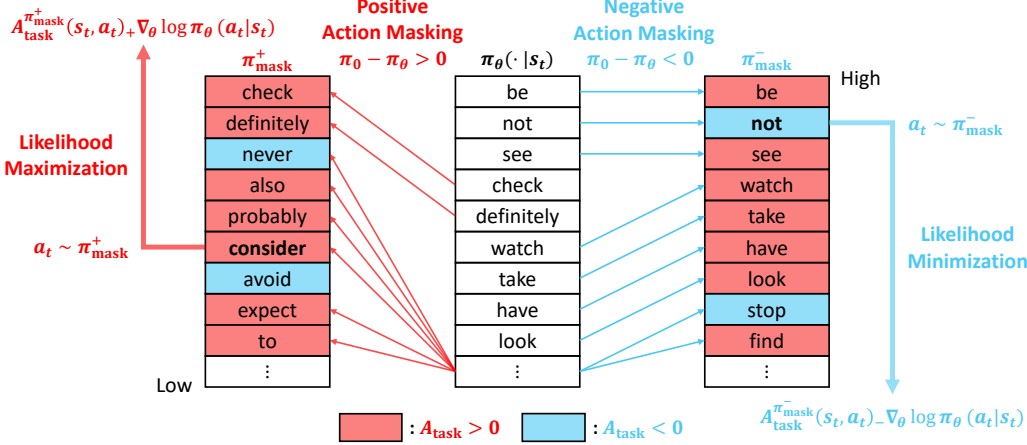

Figure 2: Illustrative example of the main mechanism of DfPO on IMDB text continuation task. The figure shows the process of sampling the action and updating the policy for the current state $s_t$. Each block is a list of the top-$k$ actions of each policy in descending order, and the color of the block indicates whether the value of advantage for the task reward is positive (red) or negative (blue). The likelihood maximization marked in red on the left corresponds to the first term in Eq. 7, and the likelihood maximization marked in blue on the right corresponds to the second term in Eq. 7.

can easily determine the samples that need to maximize or minimize the likelihood in terms of task performance by clipping the advantage function for task reward, then update the policy with likelihood maximization and minimization as follows:

$$\nabla_\theta J(\theta) = \underbrace{\mathbb{E}_{d^{\pi_{\text{mask}}^+}}\left[A^{\pi_{\text{mask}}^+}(s,a)_+ \nabla_\theta \log \pi_\theta(a|s)\right]}_{\substack{\text{Likelihood maximization for actions} \\ \text{with both positive advantages}}} + \underbrace{\mathbb{E}_{d^{\pi_{\text{mask}}^-}}\left[A^{\pi_{\text{mask}}^-}(s,a)_- \nabla_\theta \log \pi_\theta(a|s)\right]}_{\substack{\text{Likelihood minimization for actions} \\ \text{with both negative advantages}}}, \quad (7)$$

where $(\cdot)_+ = \max(\cdot, 0)$ and $(\cdot)_- = \min(\cdot, 0)$. The first term in Eq. 7 corresponds to the likelihood maximization in Table 1, and it increases the likelihood of actions with positive advantages for task reward among actions with a lower likelihood in the current policy than in the reference policy. On the other hand, the second term in Eq. 7 corresponds to the likelihood minimization in Table 1, and it decreases the likelihood of actions with negative advantages for task reward among actions with a higher likelihood in the current policy than in the reference policy. After learning through Eq. 7, the negative action-masked policy $\pi_{\text{mask}}^-$ of the updated policy $\pi_\theta$ finally consists of actions with high task performance while preserving naturalness of the generated texts. Therefore, we use the negative action-masked policy $\pi_{\text{mask}}^-(\cdot|s, \pi_\theta, \pi_0)$ as the final policy for inference. In Appendix E, we also provide a detailed explanation of how our main objective Eq. 7 was derived based on the objective of PPO.

## 3.4 DEGENERATION-FREE POLICY OPTIMIZATION

Finally, we present Degeneration-free Policy Optimization (DfPO), a new policy optimization method that can fine-tune LMs to generate texts that achieve improved downstream task scores, while preserving the naturalness of the generated texts. Figure 2 shows an illustrative example of the main mechanism of DfPO, and our algorithm can be summarized as follows: 1) action-masked policies are separately defined for the action sets that need to increase or decrease the likelihood in terms of naturalness, 2) among the action set of the positive action-masked policy, the promising action with a positive task advantage is increased and gradually moved to the action set of the negative action-masked policy, which corresponds to the likelihood maximization part on the left of Figure 2. Similarly, among the action set of the negative action-masked policy, the unpromising action with a negative task advantage is decreased and gradually moved to the action set of the positive action-

masked policy, which corresponds to the likelihood minimization part on the right of Figure 2. The main difference between DfPO and PPO is that DfPO uses KL divergence (i.e. log ratio with the initial model) to constrain action support, while PPO uses KL divergence as a reward penalty. This difference in the use of KL divergence allows DfPO improves task performance while maintaining the naturalness without additional hyperparameter search. Algorithm 1 summarizes the whole process of DfPO as the pseudocode. Since our algorithm updates the policy to maximize or minimize the likelihood of generated sentences (i.e. sentence-level policy optimization), even if it is not provided in the form of Gym-like (Brockman et al., 2016) learning environment, it can be easily applied to any dataset if only the reward function is provided. For the advantage estimation, we use Generalized Advantage Estimation (GAE) (Schulman et al., 2018), but any other advantage estimation method can be used.

---

**Algorithm 1** Degeneration-free Policy Optimization (DfPO)

---

**Input:** Training dataset $\mathcal{D} = \{(s_t^j, a_t^j, s_{t+1}^j)_{t=0}^T\}_{j=1}^N$, a policy network $\pi_\theta$ with parameter $\theta$, a value network $V_\phi$ with parameter $\phi$, a reference policy $\pi_0$

1: **for** each iteration $i$ **do**
2:     Define action-masked policies with $\pi_0$ as Eq. 5 and Eq. 6:

$$\pi_{\text{mask}}^+(\cdot|s, \pi_\theta, \pi_0) := \begin{cases} \pi_\theta(\cdot|s)/Z^+ & \text{if } \log \pi_0 - \log \pi_\theta > 0 \\ 0 & \text{otherwise} \end{cases},$$

$$\pi_{\text{mask}}^-(\cdot|s, \pi_\theta, \pi_0) := \begin{cases} \pi_\theta(\cdot|s)/Z^- & \text{if } \log \pi_0 - \log \pi_\theta < 0 \\ 0 & \text{otherwise} \end{cases}.$$

3:     Sample mini-batch of initial states $\{s_0^p\}_{p=1}^M$ and $\{s_0^n\}_{n=1}^M$ from $\mathcal{D}$
4:     Generate trajectories $\mathcal{T}^+ = \{s_t^p, a_t^p\}$ and $\mathcal{T}^- = \{s_t^n, a_t^n\}$ by running policy $\pi_{\text{mask}}^+$ and $\pi_{\text{mask}}^-$
5:     Update policy via likelihood max/minimization with clipped advantage functions as Eq. 7:

$$\arg\max_\theta \sum_{p=1}^M \sum_{t=0}^T A_\phi^{\pi_{\text{mask}}^+}(s_t^p, a_t^p)_+ \nabla_\theta \log \pi_\theta(a_t^p|s_t^p) + \sum_{n=1}^M \sum_{t=0}^T A_\phi^{\pi_{\text{mask}}^-}(s_t^n, a_t^n)_- \nabla_\theta \log \pi_\theta(a_t^n|s_t^n)$$

6:     Update the value function $V_\phi^{\pi_{\text{mask}}^+}$ and $V_\phi^{\pi_{\text{mask}}^-}$:

$$\arg\min_\phi \sum_{p=1}^M \sum_{t=0}^T (V_\phi^{\pi_{\text{mask}}^+}(s_t^p) - R(s_t^p, a_t^p))^2 + \sum_{n=1}^M \sum_{t=0}^T (V_\phi^{\pi_{\text{mask}}^-}(s_t^n) - R(s_t^n, a_t^n))^2$$

7: **end for**
**Output:** updated policy $\pi_\theta$

---

## 4 RELATED WORK

**Reinforcement learning for language models** Training a language model to improve the downstream task score can be naturally considered as an RL problem Ramamurthy et al. (2023); Snell et al. (2023); Jang et al. (2022). Reinforcement learning from human feedback (RLHF) (Christiano et al., 2017; Liang et al., 2022; Kim et al., 2023; Ouyang et al., 2022; Rafailov et al., 2023) is one of the representative successes of fine-tuning LMs through reinforcement learning. However, optimizing LMs against the reward model by using RL is yet challenging due to the *degeneration problem* which generates responses diverging from human language. Recently, as a benchmark of evaluating RL algorithms for fine-tuning language models, Ramamurthy et al. (2023) introduce (1) RL4LMs which is a modular library for optimizing LMs with RL, and (2) GRUE benchmark that is a set of language generation tasks with reward functions. Our work is based on RL4LMs and GRUE, and aims to address the degeneration problem in optimizing text generation with RL.

**Stabilizing policy gradient methods** Stabilizing policy gradient (PG) methods Peters & Schaal (2008) is essential to successfully optimize a policy, since PG methods use an *estimator* of the gradient of the expected return. TRPO Schulman et al. (2015) provides a practical algorithm by making approximations to the theoretically justified algorithm that guarantees policy improvements. PPO Schulman et al. (2017) is a simplified method from TRPO by introducing a clipped probability

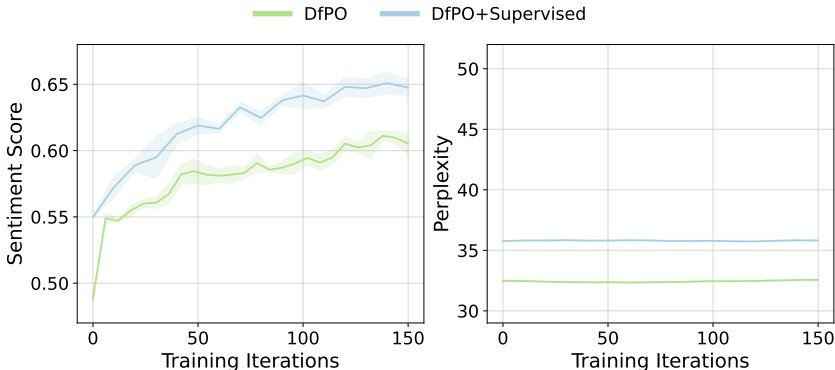

Figure 3: Experimental results of DfPO on text continuation task with IMDB dataset. DfPO and DfPO+Supervised indicate the results of DfPO that trained starting with pre-trained language model and supervised fine-tuning model, respectively. All results are averaged over 5 runs, and the shaded area represents the standard error.

ratio, while attaining the data efficiency and reliability of TRPO. Unlike these methods, NLPO Ramamurthy et al. (2023) is introduced by mainly considering text generation tasks that have much larger action space (i.e., a large number of tokens to select) than conventional decision-making tasks. NLPO mitigates the instability of policy optimization with action masking that learns to invalidate less relevant tokens. Unlike NLPO, our algorithm DfPO uses *action-masked policy* that avoids to select risky tokens that may result in the text degeneration of the original LM.

## 5 EXPERIMENTS

In this section, we show the experimental results of DfPO on generative NLP tasks included in the GRUE benchmark (Ramamurthy et al., 2023). In order to show that our algorithm improves downstream task score *while preserving the naturalness* of the generated texts, we use IMDB (text continuation) (Maas et al., 2011) and CommonGen (commonsense generation) (Lin et al., 2020) tasks as testbed which are categorized as *easily reward hackable tasks* by Ramamurthy et al. (2023). First, we show that our algorithm improves task performance while preserving the naturalness of the generated texts without a KL divergence penalty on the IMDB text continuation task. Second, we evaluate the overall performance of DfPO on the IMDB text continuation task and commonsense generation task, comparing with baseline methods including supervised learning and reinforcement learning algorithms. The details for the experimental settings can be found in Appendix A.

**Baselines** We compare the following algorithms to evaluate whether DfPO improves task performance while preserving the naturalness of the generated texts: 1) Zero Shot, a pre-trained language model without any fine-tuning of downstream tasks, 2) Supervised, a supervised fine-tuning model with datasets for downstream tasks, 3) PPO (Schulman et al., 2017), a policy gradient method that is state-of-the-art in discrete action space, 4) NLPO (Ramamurthy et al., 2023), a PPO-based policy optimization method for NLP tasks that effectively reduces the combinatorial action space with action masking. For NLPO and PPO, KL divergence from the reference policy was used as the reward penalty, and the result was obtained by hyperparameter tuning for the coefficient of the KL divergence penalty. Our goal is to improve task performance while preserving the naturalness of the generated texts without additional hyperparameter search, not to outperform PPO and NLPO obtained through hyperparameter tuning.

### 5.1 EVALUATION ON IMDB TEXT CONTINUATION TASK

We evaluate our algorithm on the IMDB text continuation task, which is one of the representative generative NLP tasks for evaluating the RL algorithms. The IMDB text continuation task aims to positively complete the movie review when given a partial review as a prompt. In this task, we use GPT-2 (117M parameters) and GPT-J (6B parameters) as a policy, and a trained sentiment classifier DistilBERT (Sanh et al., 2019) is provided as a reward function to train the RL agents and evaluate their task performance. The naturalness of the trained model is evaluated with a perplexity score.

Table 2: Experimental results on IMDB text continuation task. Cold starting indicates the results of each algorithm trained starting with the pre-trained language model (i.e. Zero Shot), and Warm starting indicates the results of each algorithm trained starting with the supervised fine-tuning model (i.e. Supervised). The results of other algorithms are from (Ramamurthy et al., 2023). All results indicate averages and standard errors over 5 independent runs.

|  | Algorithms | Sentiment Score ↑ | Perplexity ↓ |
|---|---|---|---|
| Cold Starting | Zero Shot | $0.489 \pm 0.006$ | $32.171 \pm 0.137$ |
|  | Supervised | $0.539 \pm 0.004$ | $35.472 \pm 0.074$ |
|  | PPO | $0.602 \pm 0.012$ | $33.816 \pm 0.233$ |
|  | NLPO | $0.611 \pm 0.023$ | $33.832 \pm 0.283$ |
|  | DfPO (ours) | $0.621 \pm 0.008$ | $32.531 \pm 0.099$ |
| Warm Starting | PPO+Supervised | $0.626 \pm 0.014$ | $35.049 \pm 0.347$ |
|  | NLPO+Supervised | $0.620 \pm 0.014$ | $34.816 \pm 0.340$ |
|  | DfPO+Supervised (ours) | $\mathbf{0.662 \pm 0.006}$ | $35.791 \pm 0.071$ |

Table 3: Experimental results for diversity on IMDB text continuation task. The results of other algorithms are from (Ramamurthy et al., 2023). All results indicate averages and standard errors over 5 independent runs.

| Algorithms | MSTTR | Distinct$_1$ | Distinct$_2$ | H$_1$ | H$_2$ | Unique$_1$ | Unique$_2$ |
|---|---|---|---|---|---|---|---|
| Zero Shot | $0.682 \pm 0.001$ | $0.042 \pm 0.001$ | $0.294 \pm 0.001$ | $8.656 \pm 0.004$ | $13.716 \pm 0.003$ | $5063 \pm 15$ | $47620 \pm 238$ |
| Supervised | $0.682 \pm 0.001$ | $0.047 \pm 0.001$ | $0.312 \pm 0.002$ | $8.755 \pm 0.012$ | $13.806 \pm 0.016$ | $5601 \pm 57$ | $51151 \pm 345$ |
| PPO | $0.664 \pm 0.007$ | $0.042 \pm 0.001$ | $0.278 \pm 0.005$ | $8.529 \pm 0.037$ | $13.366 \pm 0.119$ | $5108 \pm 204$ | $45158 \pm 961$ |
| PPO+Sup | $0.668 \pm 0.004$ | $0.048 \pm 0.002$ | $0.307 \pm 0.008$ | $8.704 \pm 0.053$ | $13.656 \pm 0.066$ | $5757 \pm 324$ | $50522 \pm 1514$ |
| NLPO | $0.670 \pm 0.002$ | $0.043 \pm 0.002$ | $0.286 \pm 0.006$ | $8.602 \pm 0.049$ | $13.530 \pm 0.076$ | $5179 \pm 196$ | $46294 \pm 1072$ |
| NLPO+Sup | $0.672 \pm 0.006$ | $0.048 \pm 0.002$ | $0.310 \pm 0.012$ | $8.725 \pm 0.090$ | $13.709 \pm 0.174$ | $5589 \pm 140$ | $50734 \pm 1903$ |
| DfPO | $0.710 \pm 0.008$ | $0.059 \pm 0.004$ | $0.368 \pm 0.019$ | $9.094 \pm 0.113$ | $14.387 \pm 0.203$ | $7607 \pm 603$ | $61105 \pm 3305$ |
| DfPO+Sup | $0.711 \pm 0.006$ | $0.061 \pm 0.003$ | $0.377 \pm 0.014$ | $9.153 \pm 0.087$ | $14.446 \pm 0.144$ | $7777 \pm 434$ | $62852 \pm 2474$ |

**Learning curves** Figure 3 shows the learning curves for DfPO and DfPO+Supervised with GPT-2 model that start from a pre-trained language model (i.e. Zero Shot) and supervised fine-tuning model (i.e. Supervised) as initial policies, respectively. Here, for the reference policies used when defining the action-masked policies, DfPO and DfPO+Supervised used the Zero Shot model and Supervised model same as initial policies. The results show that our algorithm DfPO successfully improves the sentiment score (i.e. task performance) while preserving their initial perplexity score (i.e. naturalness) *without additional hyperparameter search* such as the coefficient of KL penalty in PPO and NLPO. In addition, since our proposed method maximizes only the sentiment score while preserving naturalness, we can simply select the model with the best sentiment score on the validation dataset, without any ambiguity when performing model selection. Furthermore, to investigate the role of each part of DfPO, we also provide ablation studies of DfPO in Appendix C.1.

**Comparison with baselines** For the final results of DfPO, the models with the highest sentiment score on the validation dataset were selected and used for evaluation on the test dataset. Table 2 summarizes the performance of DfPO and baseline algorithms on the IMDB text continuation task. The results of other algorithms are from (Ramamurthy et al., 2023), and the results of PPO and NLPO are obtained using the coefficients of the KL penalty selected through hyperparameter search. As shown in Table 2, although the results of DfPO are obtained without additional hyperparameter search, it achieves similar performance to PPO and NLPO in optimizing sentiment scores while preserving the perplexity of initial policy (i.e. Zero Shot model for DfPO and Supervised model for DfPO+Supervised, respectively).

**Results of diversity** We also evaluate the diversity of DfPO, which is one of the most important factors when fine-tuning LMs by using reinforcement learning. Table 3 summarizes the results for diversity metrics on the IMDB text continuation task. The results show that DfPO can generate more diverse sentences than baseline algorithms for all diversity metrics. Unlike existing methods that simply explore only top-$p$ or top-$k$ actions with high probability, DfPO explores more diverse actions including both action sets with higher and lower likelihood than the reference policy. As a result, DfPO effectively optimizes task scores while maintaining the ability to generate diverse sentences through the exploration in diverse actions.

Table 4: Experimental results on generative commonsense task. The table shows experimental results on the generative commonsense task. All results indicate averages over 5 independent runs, and their standard errors are provided in Appendix due to space limit.

| Algorithms | Rouge-1 | Rouge-2 | Rouge-L | Rouge-LSum | Meteor | BLEU | BertScore | Cider | Spice |
|---|---|---|---|---|---|---|---|---|---|
| Zero Shot | 0.415 | 0.016 | 0.270 | 0.270 | 0.179 | 0.0 | 0.854 | 0.640 | 0.231 |
| Supervised | 0.503 | 0.175 | 0.411 | 0.411 | 0.309 | 0.069 | 0.929 | 1.381 | 0.443 |
| PPO+Sup | 0.540 | 0.204 | 0.436 | 0.436 | 0.329 | 0.076 | 0.930 | 1.474 | 0.447 |
| NLPO+Sup | 0.537 | 0.201 | 0.431 | 0.431 | 0.326 | 0.074 | 0.930 | 1.464 | 0.448 |
| DfPO+Sup | **0.554** | **0.221** | **0.444** | **0.443** | **0.335** | 0.079 | 0.929 | 1.455 | 0.445 |

Table 5: Experimental results for diversity on generative commonsense task. The results of other algorithms are from (Ramamurthy et al., 2023). All results indicate averages and standard errors over 5 independent runs.

| Algorithms | MSTTR | Distinct$_1$ | Distinct$_2$ | H$_1$ | H$_2$ | Unique$_1$ | Unique$_2$ |
|---|---|---|---|---|---|---|---|
| Zero Shot | 0.430 | 0.090 | 0.335 | 5.998 | 7.957 | 345 | 1964 |
| Supervised | $0.509 \pm 0.001$ | $0.101 \pm 0.001$ | $0.339 \pm 0.001$ | $6.531 \pm 0.006$ | $10.079 \pm 0.016$ | $304 \pm 7$ | $2159 \pm 25$ |
| PPO+Sup | $0.514 \pm 0.004$ | $0.105 \pm 0.002$ | $0.378 \pm 0.008$ | $6.631 \pm 0.053$ | $10.270 \pm 0.064$ | $507 \pm 17$ | $2425 \pm 73$ |
| NLPO+Sup | $0.516 \pm 0.006$ | $0.106 \pm 0.002$ | $0.377 \pm 0.008$ | $6.634 \pm 0.044$ | $10.260 \pm 0.077$ | $506 \pm 4$ | $2401 \pm 39$ |
| DfPO+Sup | $0.489 \pm 0.007$ | $0.100 \pm 0.003$ | $0.368 \pm 0.006$ | $6.549 \pm 0.015$ | $10.174 \pm 0.021$ | $471 \pm 12$ | $2376 \pm 23$ |

**Learning with large language model** We further investigate whether DfPO successfully improves downstream task score without degeneration problems, even when using a large language model as a policy. Figure 6 in Appendix C.2 shows the learning curves for DfPO with GPT-J (6B parameters) as initial and reference policy. Similar to the learning curve of DfPO with GPT-2, the results show that DfPO with a large language model successfully improves the sentiment score while preserving their initial perplexity score *without additional hyperparameter search* such as the coefficient of KL penalty in PPO and NLPO.

## 5.2 Evaluation on Commonsense Generation Task

We also evaluate our algorithm on the commonsense generation task (Lin et al., 2020), where the goal is to generate a sentence describing a scene using given concepts. In this task, we use T5 (220M parameters) as a policy model, and use a METEOR (Banerjee & Lavie, 2005) score as a reward function for training and evaluation of task performance, and BERTScore (Zhang et al., 2019) and SPICE (Anderson et al., 2016) as evaluation metrics for naturalness scores. Therefore, the goal is to successfully improve the METEOR score without degrading other task scores and naturalness scores. Table 4 and 5 summarize the results of DfPO and baseline algorithms on the commonsense generation task. As shown in Table 4, DfPO successfully improves the METEOR score while preserving other task scores and the naturalness scores. In addition, as shown in Table 5, DfPO shows similar diversity in all metrics for diversity only except MSTTR compared to other algorithms.

## 6 Conclusion

In this paper, we introduce Degeneration-free Policy Optimization (DfPO) that can fine-tune LMs to generate texts that achieve improved downstream task scores, while preserving the naturalness of the generated texts. The basic idea is to sample texts that can simultaneously optimize both the task and the naturalness rewards without interference from each other, when fine-tuning LMs. To achieve this, we develop action-masked policy with which a behavior policy can avoid selecting word tokens that potentially make policy optimization unexpected. Then, we devise a policy gradient method that separately performs likelihood maximization and minimization by using clipped advantage functions. Although DfPO requires some additional costs for inference, DfPO is much more effective than PPO and NLPO in optimizing task scores and preserving the naturalness of generated texts without additional hyperparameter search for training. As future work, we will attempt to apply DfPO to the RLHF framework with large language models. We expect that DfPO will serve as an important step towards providing a stable and robust RL method for LLM fine-tuning research.

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

# A  EXPERIMENTAL DETAILS

## A.1  TASK SPECIFICATION AND HYPERPARAMETER CONFIGURATION

Table 6 summarizes the task specifications and hyperparameter settings that we used in our experiments. For a fair comparison, we use exactly same settings of task, decoding strategy and tokenizer that used in (Ramamurthy et al., 2023). We also provide settings of hyperparameters that used in our experiments.

| | | IMDB (Text Continuation) | CommonGen (Commensense Generation) |
|---|---|---|---|
| Task Specification | task preference metric
naturalness metric | Learned Sentiment Classifier
Perplexity | METEOR
SPICE |
| Decoding | sampling
min length
max new tokens | top-$k$ ($k = 50$)
48
48 | beam search ($n = 5$)
5
20 |
| Tokenizer | padding side
truncation side
max length | left
left
64 | left
-
20 |
| Hyper-parameters | batch size
learning rate
discount factor
gae lambda | 16
0.00001
0.99
0.95 | 16
0.00001
0.99
0.95 |

Table 6: Task specification and hyperparameter configuration used in our experimental results on IMDB and CommonGen domain.

## A.2  IMPLEMENTATION DETAILS OF DFPO

We implement DfPO based on the codebase of NLPO (Ramamurthy et al., 2023), which is one of the representative RL algorithms for NLP tasks. As for the policy network, GPT-2 for the IMDB text continuation task and T5 for the generative commonsense task were used as in (Ramamurthy et al., 2023). We provide the pseudocode of our algorithm DfPO in Algorithm 1, and our code is available on `https://anonymous.4open.science/r/iclr2024-dfpo`. For the advantage function estimation, we use Generalized Advantage Estimation (GAE) (Schulman et al., 2018), but any other advantage function estimation method can be used. Since our algorithm generates sentences through action-masked policies and maximizes or minimizes the likelihood of generated sentences, we implemented our algorithm as a sentence-level policy optimization (not a word-level policy optimization).

# B    RESULTS OF NLPO WITH KL-REGULARIZATION PENALTY

We also provide the results of NLPO with KL-regularization penalty on the IMDB text continuation task. As shown in Figure 4, NLPO with KL-regularization penalty also shows very sensitive performance on both sentiment score and perplexity to hyperparameter.

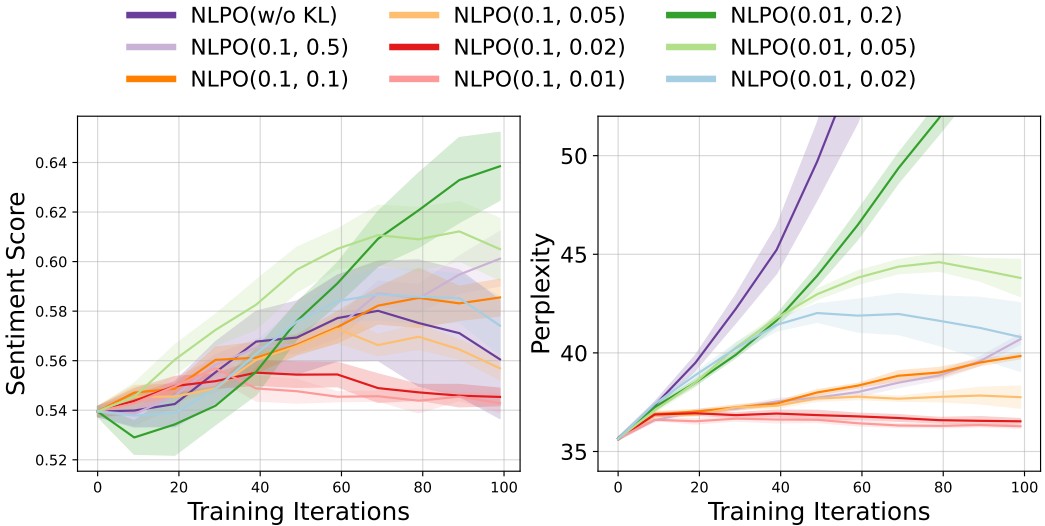

Figure 4: Averaged learning curve over 5 runs of NLPO with KL-regularization penalty on IMDB text continuation task for varying KL coefficient and target KL. NLPO ($\beta$, $KL_{target}$) indicates the NLPO that considers the KL-regularization as a reward penalty with KL coefficient $\beta$ and target KL. The goal of the IMDB text continuation task is to learn a policy that maximizes the sentiment score (i.e. task reward) while preserving the perplexity (i.e. naturalness) of the initial policy. However, as shown in the results, NLPO with KL-regularization penalty shows very sensitive performance on both sentiment score and perplexity to hyperparameter.

## C  ADDITIONAL EXPERIMENTAL RESULTS

### C.1  ABLATION STUDY

To study the role of each part of the objective, we provide ablation study results of DfPO on the IMDB text continuation task. We compare the results of methods trained with the four types of objectives (Policy gradient, Likelihood Maximization, Likelihood Minimization, and DfPO), and the details of each model are provided below. As shown in Figure 5, the naive policy gradient (PG), similar to PPO without the KL regularization penalty in Figure 1, improves task performance but the perplexity diverges (i.e. deteriorates the naturalness of generated texts). In the case of updating the policy with only likelihood minimization, it fails to improve the task performance and also preserve the naturalness of generated texts. In addition, when updating the policy with only likelihood maximization, the perplexity does not diverge, but the task performance is improved relatively low compared to DfPO. Therefore, each part of the objective in DfPO is necessary to successfully improve the task performance while preserving the naturalness of generated texts.

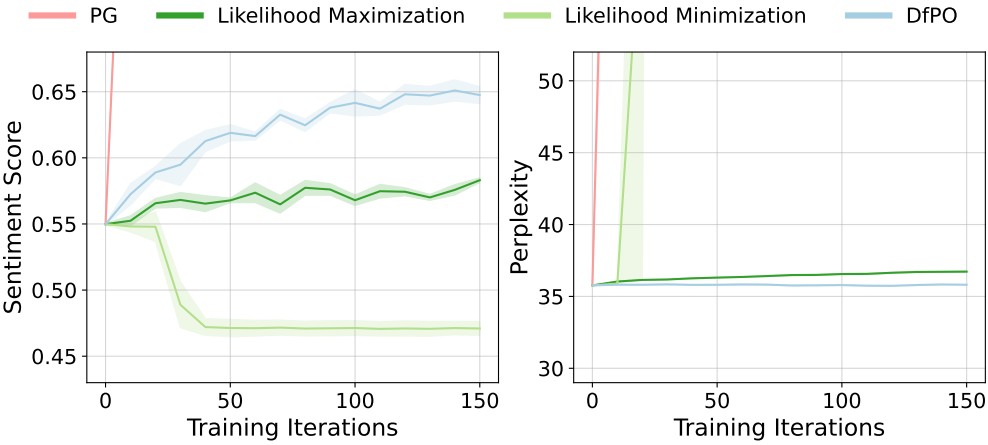

Figure 5: Ablation study results of DfPO on text continuation task with IMDB dataset. All results are averaged over 5 runs, and the shaded area represents the standard error.

**Degeneration-free Policy Optimization (DfPO)**

To show the role of each part of the objective, we compare the result of DfPO trained with the following objective, which is exactly the same as the result of DfPO+Supervised in Figure 3:

$$\nabla_\theta J(\theta) = \underbrace{\mathbb{E}_{a \sim \pi^+_{\text{mask}}(s)}\left[A^{\pi^+_{\text{mask}}}_{\text{task}}(s,a)_+ \nabla_\theta \log \pi_\theta(a|s)\right]}_{\substack{\text{Likelihood maximization for actions} \\ \text{with both positive advantages}}} + \underbrace{\mathbb{E}_{a \sim \pi^-_{\text{mask}}(s)}\left[A^{\pi^-_{\text{mask}}}_{\text{task}}(s,a)_- \nabla_\theta \log \pi_\theta(a|s)\right]}_{\substack{\text{Likelihood minimization for actions} \\ \text{with both negative advantages}}}.$$

(8)

**Likelihood Maximization**

First, we compare the results of updating the policy through the only likelihood maximization, which corresponds to the first term of the main objective of DfPO:

$$\nabla_\theta J(\theta) = \underbrace{\mathbb{E}_{a \sim \pi^+_{\text{mask}}(s)}\left[A^{\pi^+_{\text{mask}}}_{\text{task}}(s,a)_+ \nabla_\theta \log \pi_\theta(a|s)\right]}_{\substack{\text{Likelihood maximization for actions} \\ \text{with both positive advantages}}}.$$

(9)

**Likelihood Minimization**

Second, we compare the results of updating the policy through the only likelihood minimization, which corresponds to the second term of the main objective of DfPO:

$$\nabla_\theta J(\theta) = \underbrace{\mathbb{E}_{a \sim \pi_{\text{mask}}^-(s)} \left[ A_{\text{task}}^{\pi_{\text{mask}}^-}(s, a)_- \nabla_\theta \log \pi_\theta(a|s) \right]}_{\substack{\text{Likelihood minimization for actions} \\ \text{with both negative advantages}}}. \tag{10}$$

**Policy Gradient (PG)**

We also compare the result of naive policy gradient update without action-masked policies and advantage clipping as follows:

$$\nabla_\theta J(\theta) = \mathbb{E}_{a \sim \pi_\theta(s)} \left[ A^{\pi_\theta}(s, a) \nabla_\theta \log \pi_\theta(a|s) \right]. \tag{11}$$

## C.2 EXPERIMENTAL RESULTS WITH LARGE LANGUAGE MODEL

We also provide the result of DfPO with a large language model on the IMDB text continuation task. For the large language model, we use GPT-J (6B parameters) as initial and reference policies. As shown in Figure 6, DfPO can improve task scores while preserving the naturalness of the generated texts, even when using a large language model (GPT-J-6B) as a policy.

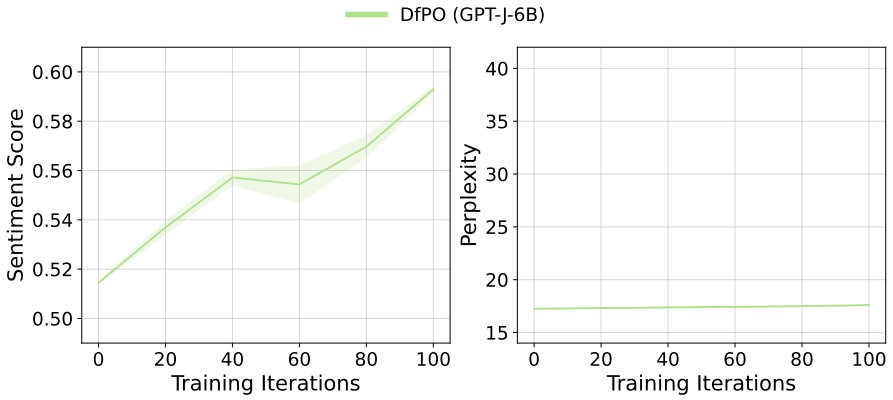

Figure 6: Experimental results of DfPO with large language model on text continuation task with IMDB dataset. The plots show the results of DfPO that was trained starting with GPT-J (6B parameters) model. All results are averaged over 5 runs, and the shaded area represents the standard error.

## C.3 EXPERIMENTAL RESULTS ON GENERATIVE COMMONSENSE TASK WITH STANDARD ERRORS

Due to the space limit of main paper, we provide the whole results of Table 4, including the standard errors.

| Algorithms | Rouge-1 | Rouge-2 | Rouge-L | Rouge-LSum | Meteor | BLEU | BertScore | Cider | Spice |
|---|---|---|---|---|---|---|---|---|---|
| Zero Shot | 0.415 | 0.016 | 0.270 | 0.270 | 0.179 | 0.0 | 0.854 | 0.640 | 0.231 |
| Supervised | $0.503 \pm 0.001$ | $0.175 \pm 0.001$ | $0.411 \pm 0.001$ | $0.411 \pm 0.001$ | $0.309 \pm 0.001$ | $0.069 \pm 0.001$ | $0.929 \pm 0.000$ | $1.381 \pm 0.011$ | $0.443 \pm 0.001$ |
| PPO+Sup | $0.540 \pm 0.005$ | $0.204 \pm 0.005$ | $0.436 \pm 0.004$ | $0.436 \pm 0.004$ | $0.329 \pm 0.003$ | $0.076 \pm 0.003$ | $0.930 \pm 0.001$ | $1.474 \pm 0.022$ | $0.447 \pm 0.004$ |
| NLPO+Sup | $0.537 \pm 0.003$ | $0.201 \pm 0.004$ | $0.431 \pm 0.002$ | $0.431 \pm 0.002$ | $0.326 \pm 0.002$ | $0.074 \pm 0.003$ | $0.930 \pm 0.000$ | $1.464 \pm 0.025$ | $0.448 \pm 0.002$ |
| DfPO+Sup | $\mathbf{0.554 \pm 0.002}$ | $\mathbf{0.221 \pm 0.002}$ | $\mathbf{0.444 \pm 0.001}$ | $\mathbf{0.443 \pm 0.001}$ | $\mathbf{0.335 \pm 0.001}$ | $0.079 \pm 0.001$ | $0.929 \pm 0.001$ | $1.455 \pm 0.027$ | $0.445 \pm 0.002$ |

Table 7: Experimental results on generative commonsense task. The table shows experimental results on the generative commonsense task with averages and standard errors. All results indicate averages and standard errors over 5 independent runs.

## C.4 LEARNING CURVE OF DFPO ON COMMONSENSE GENERATION TASK

We also provide the learning curve of DfPO on the commonsense generation task. As shown in Figure C.3, DfPO can improve task scores (i.e. METEOR) while preserving the naturalness (i.e. Perplexity) of the generated texts.

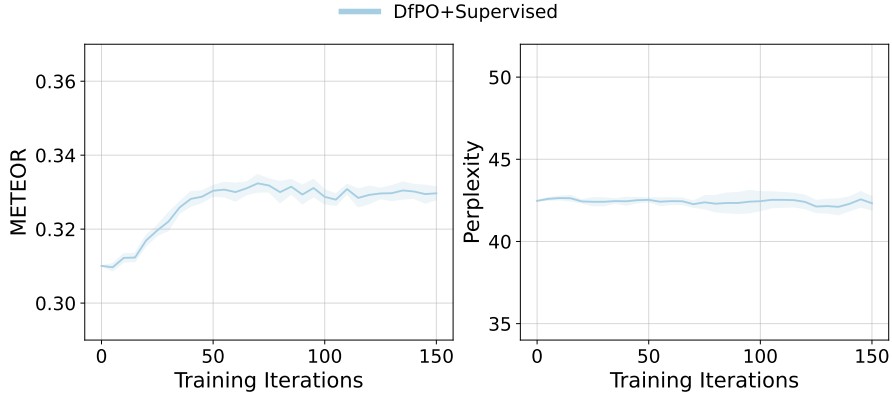

Figure 7: Experimental results of DfPO on commonsense generation task. All results are averaged over 5 runs, and the shaded area represents the standard error.

## C.5 RESULTS OF PPO WITH SFT

We conducted additional experiments for PPO+SFT, where the learning objective is aggregated with PPO and supervised fine-tuning objectives. We provide the results of PPO+SFT with KL-regularization penalty on the IMDB text continuation task. As shown in Figure C.3, PPO+SFT with KL-regularization penalty also shows very sensitive performance on both sentiment score and perplexity to hyperparameter.

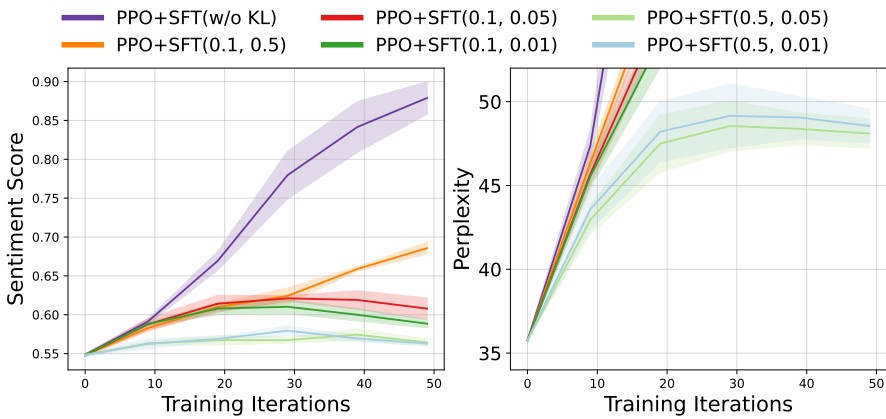

Figure 8: Averaged learning curve over 5 runs of PPO+SFT with KL-regularization penalty on IMDB text continuation task for varying KL coefficient and target KL. PPO+SFT ($\beta$, KL$_{\text{target}}$) indicates the PPO+SFT that considers the KL-regularization as a reward penalty with KL coefficient $\beta$ and target KL. The goal of the IMDB text continuation task is to learn a policy that maximizes the sentiment score (i.e. task reward) while preserving the perplexity (i.e. naturalness) of the initial policy. However, as shown in the results, PPO+SFT with KL-regularization penalty shows very sensitive performance on both sentiment score and perplexity to hyperparameter.

## C.6 RESULTS WITH VARYING POLICY FOR INFERENCE

To study the role of negative action-masked policy for inference, we provide additional experimental results of DfPO on the IMDB text continuation task. We compare the results of DfPO with three types of policies for inference ($\pi_\theta$, positive action-masked policy $\pi_{\text{mask}}^+$, and negative action-masked policy $\pi_{\text{mask}}^-$). As shown in Figure C.3, we have experimentally observed that the task performance deteriorates when inference is performed with other policies ($\pi_\theta$ and positive action-masked policy $\pi_{\text{mask}}^+$) rather than the negative action-masked policy $\pi_{\text{mask}}^-$.

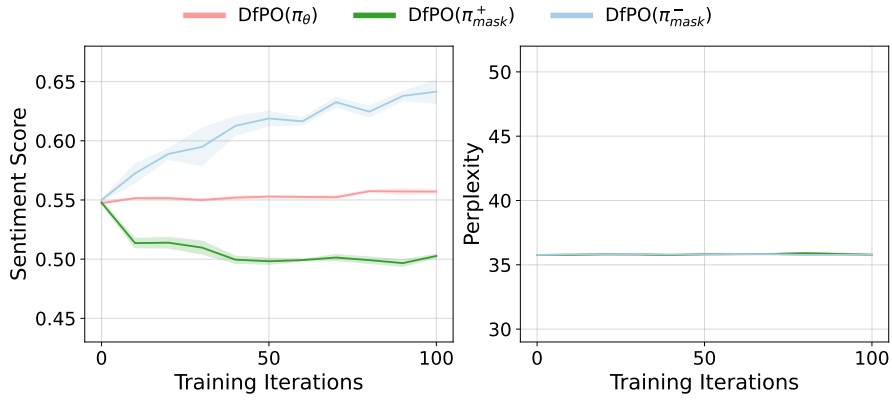

Figure 9: Experimental results of DfPO with varying policy for inference. All results are averaged over 5 runs, and the shaded area represents the standard error.

# D  QUALITATIVE RESULTS

In this section, we demonstrate through qualitative examples that DfPO improves task performance while maintaining the naturalness of generated texts. We provide generation results from DfPO and baseline algorithms. We selected the same prompts as used in (Ramamurthy et al., 2023), and the generation results of baseline algorithms are from (Ramamurthy et al., 2023). As shown in the examples, the generation results of DfPO show maintaining naturalness, which does not diverge from human language. On the other hand, PPO-no-KL and NLPO-no-KL, which do not use the KL regularization penalty, show results with poor naturalness of generated texts.

**Sample** 1

**Prompt:**
```
A man and his wife are not getting along because of the wife's jealousy
and drinking problems.  When the wife disappears, her sister begins a
vigorous search involving the
```

**References:**
```
local police, state government, and a television crime show.  What she
finds out stuns the entire community.  Good drama with lots of fine
performances; based on a true story
```

**Zero-Shot:**
```
husband's sister, which leads several men to a remote cabin and then
a boat.  The wife and her husband then start to see the same things.
Finally, a strange woman is driving out to her cabin and discovers the
husband's disappearance but
```

**Supervised:**
```
amily tree.  The widower, now widowed, returns, the daughter, daughter
is not pleased when his wife is mistaken for one of her sons, and the
patriarch, then a wealthy man, falls victim to a jealous husband.  The
```

**PPO:**
```
police and the local public.  At the end, the woman is killed and the
wife is found naked and with a red head tattooed on her back.  This is
part of a special feature called"On the Road, Hidden" which was
```

**PPO-no-KL:**
```
police and other local governments.  Written by G. Scott Fitzgerald, who
also stars in this production, "A Very Merry Christmas" is a wonderful
family movie with a wonderful soundtrack and wonderful characters.  It
brings home the true beauty of small-
```

**NLPO:**
```
mother while the brother goes under.  After all, his sister, who was
apparently kidnapped but who turns out to be a man, is no longer in her
place.  If a film that was made in the 1950s
```

**NLPO-no-KL:**
```
friends of the wife, and also the friend of the wife of the woman that
is missing, and eventually he begins to find that she has gone to Germany
and he has made a very good friend.  There is a good scene between John
and
```

**DfPO:**
```
town of Berenner, California.  This is an excellent film, and it was made
one year before George Cloorman (Stick With Her).  It's as if George was
already aware of the historical context
```

**Sample** 2

**Prompt:**
A very credible and unsettling movie portraying the effects of war on
the general public, Shame gives us a chilling understanding oh Ingmar
Bergman's beliefs.  The absence of color and sound (as in a soundtrack)
helps to give the film a more realistic feeling.  There is no soft or
hard lighting or a variety of camera angles to give the story a charming
or dramatic effect.  Everything is very simply portrayed as it would be
in reality.  At times the storyline was confusing, but perhaps in order
to give a sense of the characters' frame of mind how they too do now know
what is happening or why.  The simplicity of the

**References:**
film made it somewhat boring, but I could understand how the director is
trying not to glamorize the story and points.  He depicted the harsh and
unsettling events and effects of war.  In the beginning the characters
were introduced and shown as content with the lighter settings

**Zero-Shot:**
camera work is at once apparent and fascinating in the manner in which
it affects the way things are portrayed.  In other films these two have
been referred to from time to time in a form and manner more suited to
the screen.  The film shows

**Supervised:**
"Dakota Clan" and his sister, the two become friends with Ted Masters who
soon finds the chance to fly back to their home land and go back to save
his brother's life and his family.  The

**PPO:**
characters relationships is so engrossing and the great and unique story
presented by each pair of brothers and sisters creates a truly gripping
and enjoyable experience.  To put it simply, I strongly recommend to
everyone who plays the role to buy this DVD!!!

**PPO-no-KL:**
movie helps it to flow along in a believable way.  Rating 4 out of 5
Great Movie!  I highly recommend these movies!  I highly recommend these
movies!  Rating 5 out of 5 Great Movie!  I highly recommend these movies!
Rating 10 of"

**NLPO:**
novel makes it one of the first books I have read by a British actor.  It
is a fun way for someone who likes their books, like me, to sit through a
novel.  If you like to read novels, check it out.

**NLPO-no-KL:**
setting makes the final product very enjoyable to watch.  Highly
recommended!!    Michael Bowerman, SPSA Film Festival Host, SPSA, WA
(December 17, 2006), 8/17/2006 Great photography, good acting,

**DfPO:**
screenplay which follows that, makes for an excellent point, by telling
this to the general public at hand with an all-merse narration and sound.
The scene with Sgt Columbo fighting for our pride is actually excellent
and

**Sample** 3

**Prompt:**
Shakespearean language is wild and rambling, saturated in multiple meanings, word play and metaphor.  To be understood it must be wrangled and tamed by an actor with the strength and knowledge to do so.  When an actor fails, the words pour forth in a torrent of incomprehensible words, but when he succeeds, the English language springs to life with an immediacy and vibrancy that takes your breath away.  Al Pacino is one such actor, and here displays an incredible level of clarity and control that, were there any justice,

**References:**
would sweep every award in the offering.  He meets the challenge of presenting Shylock head on, and delivers an extraordinarily subtle and nuanced performance.  It would be a crime if we never got the opportunity to see what he does with King Lear.

The supporting cast

**Zero-Shot:**
might still be seen in the world today.  This movie, like "Vacation", is an effort at telling the tale of two families from different races.  One is African but never grew up the other.  One of them, a man

**Supervised:**
his performance (so far) would seem mere shadow.  He is truly in the middle of a movie, and this film is one of those films where he can be convincing in it (and his trademark acting, as you can see in the

**PPO:**
would have made him an instant-classic.  This movie captures the spirit of childhood and has everyone in the world watching it.  It's a love story and it's really heartbreaking.  And at the same time, it's a classic.  It

**PPO-no-KL:**
will bring victory to the fight against hatred and extremism.  Al Pacino is a very beautiful, well executed actor who brings an absolute intensity and sensibility to his portrayal of love and loyalty.  A must see! Recommended!  Also, an Excellent

**NLPO:**
I would watch and enjoy.  7/10, Oscar Wilde is always a pleasure to see. A must see.  Highly recommend.  Highly recommend!  One of my top ten films of all time.  A must see!!

**NLPO-no-KL:**
the whole film would have fallen to the same fate, just as it did just a couple hundred years ago.  Don't miss it.  It's a real classic.  Highly Recommended.  * outta five stars for it!

**DfPO:**
hardly could be conceived, couldn´ve only appear to be a dream.  His acting speaks for an incredible combination of passion and fear; his portrayal of Shakespeare that comes from passion is mesmerishing.  His accent is reminiscent of the softening with

# E    COMPARISON WITH PPO

Before starting the derivation, we define the following stationary distributions:

$$d^\pi(s) := (1 - \gamma) \sum_{t=0}^\infty \gamma^t \Pr(s_t = s),$$

$$d^\pi(s, a) := (1 - \gamma) \sum_{t=0}^\infty \gamma^t \Pr(s_t = s, a_t = a),$$

where $s_0 \sim p_0$ and the actions are chosen according to $\pi$. In addition, the policy gradient objective can be formulated as

$$\nabla_\theta J_{\text{PG}}(\theta) = \mathbb{E}_{(s,a)\sim d^\pi}[A^\pi(s, a)\nabla_\theta \log \pi_\theta(a|s)].$$

## E.1    PPO OBJECTIVE

From the policy gradient objective, TRPO and PPO use the following gradient of the surrogate objective:

$$\nabla_\theta J_{\text{PG}}(\theta) \approx \mathbb{E}_{s\sim d^{\pi_{\text{old}}}, a\sim\pi(\cdot|s)}[A^{\pi_{\text{old}}}(s, a)\nabla_\theta \log \pi_\theta(a|s)]$$

$$= \mathbb{E}_{(s,a)\sim d^{\pi_{\text{old}}}}\left[A^{\pi_{\text{old}}}(s, a)\frac{\pi_\theta(a|s)\nabla_\theta \log \pi_\theta(a|s)}{\pi_{\text{old}}(a|s)}\right]$$

$$= \nabla_\theta \mathbb{E}_{(s,a)\sim d^{\pi_{\text{old}}}}\left[A^{\pi_{\text{old}}}(s, a)w_{\theta,\text{old}}(s, a)\right] =: \nabla_\theta \hat{J}_{\text{PG}}(\theta),$$

where $w_{\theta,\text{old}}(s, a) := \frac{\pi_\theta(a|s)}{\pi_{\text{old}}(a|s)}$.

To stabilize this objective, PPO modifies the surrogate objective to penalize deviations of $\pi_\theta(s, a)$ from $\pi_{\text{old}}(s, a)$ (Schulman et al., 2017):

$$J_{\text{PPO}}(\theta) = \mathbb{E}_{(s,a)\sim d^{\pi_{\text{old}}}}\left[\min\{A^{\pi_{\text{old}}}(s, a)w_{\theta,\text{old}}(s, a), A^{\pi_{\text{old}}}(s, a)\text{clip}(w_{\theta,\text{old}}(s, a), 1 - \epsilon, 1 + \epsilon)\}\right]$$

for a hyperparameter $\epsilon > 0$. For two arbitrary variables $u$ and $A$, we define following two functions:

$$f(w, A, \epsilon) := \min\{Aw, A \cdot \text{clip}(w, 1 - \epsilon, 1 + \epsilon)\},$$

$$g(w, A, \epsilon) := \{\mathbb{I}(A \geq 0, w < 1 + \epsilon) + \mathbb{I}(A < 0, w > 1 - \epsilon)\},$$

where $\mathbb{I}$ is a indicator function:

$$\mathbb{I}(cond) = \begin{cases} 1 & \text{if } cond = \texttt{True} \\ 0 & \text{otherwise} \end{cases}.$$

Then, the following equations hold:

$$\nabla_\theta J_{\text{PPO}}(\theta) = \nabla_\theta \mathbb{E}_{(s,a)\sim d^{\pi_{\text{old}}}}\left[f(w_{\theta,\text{old}}(s, a), A^{\pi_{\text{old}}}(s, a), \epsilon)\right]$$

$$= \nabla_\theta \mathbb{E}_{(s,a)\sim d^{\pi_{\text{old}}}}\left[g(w_{\theta,\text{old}}(s, a), A^{\pi_{\text{old}}}(s, a), \epsilon)A^{\pi_{\text{old}}}(s, a)w_{\theta,\text{old}}(s, a)\right]$$

$$= \nabla_\theta \mathbb{E}_{s\sim d^{\pi_{\text{old}}}, a\sim\pi(\cdot|s)}\left[g(w_{\theta,\text{old}}(s, a), A^{\pi_{\text{old}}}(s, a), \epsilon)A^{\pi_{\text{old}}}(s, a)\right],$$

and consequently, the gradient of $J_{\text{PPO}}(\theta)$ is equivalent to the gradient of the following objective:

$$\hat{J}_{\text{PPO}}(\theta) := \mathbb{E}_{s\sim d^{\pi_{\text{old}}}, a\sim\pi(\cdot|s)}\left[g(w_{\theta,\text{old}}(s, a), A^{\pi_{\text{old}}}(s, a), \epsilon)A^{\pi_{\text{old}}}(s, a)\right].$$

Compared to the original surrogate objective $\hat{J}_{\text{PG}}(\theta)$, $g(w_{\theta,\text{old}}(s, a), A^{\pi_{\text{old}}}(s, a), \epsilon)$ is added to prevent deviations of $\pi_\theta(s, a)$ too far from $\pi_{\text{old}}(s, a)$.

## E.2 DFPO OBJECTIVE

In NLP tasks, when we have a reward model, the environment interaction is not too expensive because generating an action sequence from $\pi$ is equivalent to generating a trajectory from $\pi$. Thus, we will use policy gradient objective instead of surrogate objective. In addition, we want to penalize deviations of $\pi_\theta$ from $\pi_0$, instead of $\pi_{\text{old}}$. Hence, similar to the original PPO objective, incorporating $g(w_{\theta,0}(s,a), A^\pi(s,a), \epsilon)$ into the original policy gradient objective $J_{\text{PG}}(\theta)$ can be employed to avoid substantial deviations of $\pi_\theta(s,a)$ from $\pi_0(s,a)$:

$$J_{\text{PG}}(\theta) = \mathbb{E}_{(s,a)\sim d^\pi}[A^\pi(s,a)]$$
$$\approx \mathbb{E}_{(s,a)\sim d^\pi}[g(w_{\theta,0}(s,a), A^\pi(s,a), \epsilon) A^\pi(s,a)] =: \tilde{J}_{\text{PG}}(\theta)$$

where $w_{\theta,0}(s,a) := \frac{\pi_\theta(a|s)}{\pi_0(a|s)}$. We set $\epsilon = 0$ and approximate the gradient of $\tilde{J}_{\text{PG}}(\theta)$ as follows:

$$
\begin{aligned}
\nabla_\theta \tilde{J}_{\text{PG}}(\theta) =& \mathbb{E}_{(s,a)\sim d^\pi}[\{\mathbb{I}(A^\pi(s,a) \geq 0, w_{\theta,0}(s,a) < 1) \\
& \quad + \mathbb{I}(A^\pi(s,a) < 0, w_{\theta,0}(s,a) > 1)\} A^\pi(s,a) \nabla_\theta \log \pi_\theta(a|s)] \\
=& \mathbb{E}_{(s,a)\sim d^\pi}[\mathbb{I}(w_{\theta,0}(s,a) < 1) A^\pi(s,a)_+ \nabla_\theta \log \pi_\theta(a|s)] \\
& + \mathbb{E}_{(s,a)\sim d^\pi}[\mathbb{I}(w_{\theta,0}(s,a) > 1) A^\pi(s,a)_- \nabla_\theta \log \pi_\theta(a|s)] \\
\approx& \mathbb{E}_{(s,a)\sim d^{\pi_{\text{mask}}^+}}[A^{\pi_{\text{mask}}^+}(s,a)_+ \nabla_\theta \log \pi_\theta(a|s)] + \mathbb{E}_{(s,a)\sim d^{\pi_{\text{mask}}^-}}[A^{\pi_{\text{mask}}^-}(s,a)_- \nabla_\theta \log \pi_\theta(a|s)].
\end{aligned}
$$

$$(12)$$
$$(13)$$

The last term Eq. (13) is equivalent to the DfPO objective in Eq. (7), which successfully improves the task performance while preserving the naturalness of generated texts. We experimentally observed that, unlike when updating the policy with Eq. 12, the policy is successfully improved without degeneration issue when updating with Eq. 13.

# F RESULTS OF DPO

We additionally provide the results of Direct Preference Optimization (DPO) (Rafailov et al., 2023), which is one of the recent representative RLHF algorithms.

## F.1 EXPERIMENTAL SETTINGS OF DPO

We conducted additional experiments for DPO on the IMDB text continuation task. In contrast to the online RL setting given the reward function considered in this paper, DPO assumes an offline RL setting given a pairwise dataset. For the offline pairwise dataset, we used prefixes from the IMDB training dataset as prompts, then we sampled 2 completions for 20000 prompts and created preference pairs for each prefix using the ground-truth reward model.

## F.2 EXPERIMENTAL RESULTS OF DPO

We provide the results of DPO with various hyperparameters for the KL-regularization penalty (i.e. $\beta$). As shown in Figure D, DPO with KL-regularization penalty also shows very sensitive performance on both sentiment score and perplexity to hyperparameter. We also provide the perplexity-reward frontier for DPO. Figure D shows that DfPO achieves better sentiment score performance than DPO for the same perplexity.

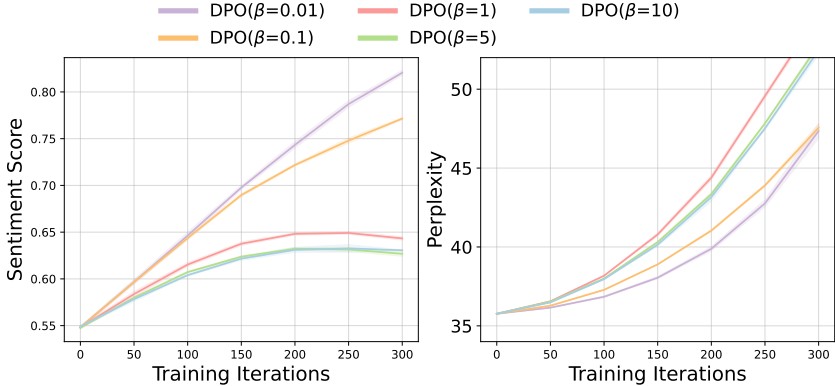

Figure 10: Averaged learning curve over 5 runs of DPO on IMDB text continuation task for varying KL coefficient $\beta$. DPO($\beta$) indicates the DPO that considers the KL-regularization as a reward penalty with KL coefficient $\beta$. As shown in the results, DPO shows very sensitive performance on both sentiment score and perplexity to hyperparameter.

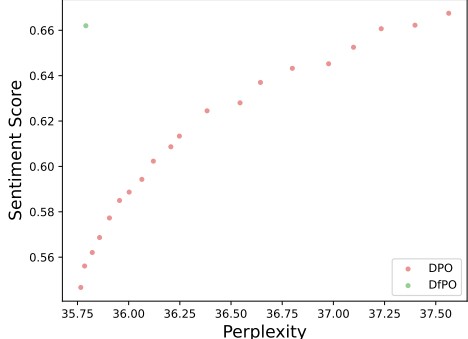

Figure 11: Perplexity-Reward frontier for DPO. The $x$-axis represents the perplexity score and the $y$-axis represents the sentiment score.

