# OpenReview forum: "DfPO: Degeneration-free Policy Optimization via Action Masking in Natural Language Action Spaces"
_ICLR.cc/2024/Conference — Submitted to ICLR 2024_

### Official Review · Reviewer_6zCc · 2023-10-29

**Soundness:** 2 fair
**Presentation:** 2 fair
**Contribution:** 3 good
**Rating:** 5
**Confidence:** 4

**Summary:**

The paper proposes an RL algorithm to address the problem of naturalness degeneration in LMs, which often occurs during RL finetuning. The proposed algorithm employs action-masked policy, a behavior policy in which actions (i.e. tokens) that potentially cause the degeneration are masked out by leveraging reference policy. This allows for separate likelihood maximization and minimization for rollout samples during learning. The authors observed gains in two natural language generation tasks.

**Strengths:**

- The motivation is intuitive. I like the interpretation that decomposes advantage function of PPO into task-specific advantage and advantage for naturalness, as outlined in the section 3.1.
- The idea of action masked policy that leverages the reference policy is novel.
- The paper is well-organized.

**Weaknesses:**

- The current version of the paper lacks some important explanations such as:
    1. When and how do the samples in the red area of Table.1 become undesired samples?
    2. How does the action-masked policy lead to better text generation than PPO?
    3. How can it ensure that the masked actions do not contribute to enhancing the task score while maintaining naturalness?
    4. How efficient is DfPO in terms of sample efficiency for learning?
- DfPO needs feed forwarding for reference policy to obtain negative action-masked policy at inference time. This demands additional resources and computation.
- The benefits from the method are limited, especially in CommonGen, in terms of naturalness and diversity. Breath of results is also limited as the authors consider only two tasks.

**Questions:**

- Reward functions we use to finetune LLMs are usually designed to consider the naturalness. In this case, the samples that have the opposite direction for improving task performance and naturalness may not be crucial to the degeneration. Thoughts?
- How does DfPO perform when it does not use negative action-masked policy during inference?
- Mixing PPO loss and SFT loss (from the same batch) is a basic way to mitigate the ‘tax’ like the naturalness degeneration in RL finetuning for language generation . This baseline should be included in this work.
- The table caption should be positioned at the top of the table.
- Use \citep after a name of a method (e.g., PPO \citep{…}) instead of \citet

---

> ### Author Response · Authors · 2023-11-20
> **Response to Reviewer 6zCc**
>
> Thank you for your constructive feedback and comments. Please feel free to ask any additional follow-up questions.
>
> **[Responses to Weaknessess]**
>
> **W1-1) When and how do the samples in the red area of Table.1 become undesired samples?**
>
> When samples in the red area are used for policy update through likelihood max/minimization (i.e. policy gradient), they become undesired samples. For example, in the case of samples in the red area at the top right of Table 1 (i.e. $\pi_{0}(a|s) > \pi_{\theta}(a|s)$), it is okay to perform likelihood maximization from a naturalness perspective, but if we perform it, task performance will deteriorate because the advantage for the task reward is negative (i.e. $A_{\text{task}}^{\pi_{\theta}}(s,a) < 0$). We added this explanation to **Section 3.1** in the revised paper.
>
> **W1-2) How does the action-masked policy lead to better text generation than PPO?**
>
> The main difference between DfPO and PPO is that DfPO uses KL divergence (i.e. log ratio with the initial model) to **constrain action support**, while PPO uses KL divergence as a **reward penalty**. This difference in the use of KL divergence allows DfPO improves task performance while maintaining the naturalness without additional hyperparameter search. We explicitly added this explanation to **Section 3.4** in the revised paper.
>
> **W1-3) How can it ensure that the masked actions do not contribute to enhancing the task score while maintaining naturalness?**
>
> Since masked actions are not used in policy update (i.e. likelihood max/minimization), they do not contribute/disturb to enhancing the task score while maintaining naturalness.
>
> **W1-4) How efficient is DfPO in terms of sample efficiency for learning?**
>
> As can be seen in Figure 1, PPO shows very different learning curves depending on hyperparameter $\beta$, so sample efficiency for learning cannot cannot be directly compared. However, it has a similar sample efficiency as PPO with a high value of $\beta$.
>
> **W2) DfPO needs feed forwarding for reference policy to obtain negative action-masked policy at inference time. This demands additional resources and computation.**
>
> It is correct that DfPO needs more feed forwarding for reference policy at inference time than other algorithms, and we explicitly added this to **Section 6** in the revised paper as a limitation.
>
> **W3) The benefits from the method are limited, especially in CommonGen, in terms of naturalness and diversity. Breath of results is also limited as the authors consider only two tasks.**
>
> In the Commonsense Generation task, we also evaluated naturalness (BertScore and Spice in Table 4) and diversity (all metrics in Table 5). As shown in the Table 4 and 5, DfPO successfully improves the METEOR score while preserving the naturalness scores and diversity.
>
> **[Responses to Questions]**
>
> **Q1) Reward functions we use to finetune LLMs are usually designed to consider the naturalness. In this case, the samples that have the opposite direction for improving task performance and naturalness may not be crucial to the degeneration. Thoughts?**
>
> Naturalness is also **implicitly** considered in the reward functions we use to finetune LLMs. However, it more directly considers the quality evaluation of the answer to the task rather than naturalness, and does not perfectly consider naturalness. Since the reward model does not perfectly consider naturalness, degeneration issues through reward hacking inevitably occur, and therefore all RLHF algorithms use the KL-divergence penalty even though they use a reward model that implicitly considers naturalness.
>
> **Q2) How does DfPO perform when it does not use negative action-masked policy during inference?**
>
> We have experimentally observed that the task performance deteriorates when inference is performed with other policies ($\pi_\theta$ and positive action-masked policy) rather than the negative action-masked policy. We added this result to **Appendix C.6** in the revised paper.
>
> **Q3) Mixing PPO loss and SFT loss (from the same batch) is a basic way to mitigate the ‘tax’ like the naturalness degeneration in RL finetuning for language generation . This baseline should be included in this work.**
>
> Thank you for suggesting the additional baseline that mixes PPO loss and SFT loss. We additionally conducted the experiment for PPO+SFT (i.e. mixing PPO loss and SFT loss), and added the results to **Appendix C.5** in the revised paper. The results show that even though it was trained with a loss that is a combination of PPO loss and SFT loss, it is still vulnerable to degeneration issues.
>
> **Q4 and Q5) The table caption should be positioned at the top of the table. & Use \citep after a name of a method (e.g., PPO \citep{…}) instead of \citet**
>
> Thank you for your feedback. We revised our manuscript based on your comments.

---

> ### Author Response · Authors · 2023-11-22
> **Gentle Reminder**
>
> Dear Reviewer 6zCc,
>
> Thanks again for your constructive feedback and comments. As the discussion period is soon to end, we would like to know if your questions and concerns have been resolved by our response and revision. We would be grateful if you could offer feedback on our rebuttal, and if your questions and concerns have been resolved by our rebuttal, we hope you update your rating accordingly. We appreciate the reviewer's insightful and constructive comments once again.
>
> Best,
>
> Authors

---

### Official Review · Reviewer_99mP · 2023-11-01

**Soundness:** 2 fair
**Presentation:** 2 fair
**Contribution:** 2 fair
**Rating:** 5
**Confidence:** 4

**Summary:**

This paper introduces a method to maintain the generation distribution (in place of the KL penalty in PPO) during the RL procedure. The basic idea is to apply RL optimization through likelihood maximization and minimization to samples only if the direction is also towards the reference policy. Experimental results on GRUE benchmark show that the proposed method outperformed PPO and NLPO.

**Strengths:**

Experiments on the GRUE benchmark show that DfPO indeed get better the downstream task scores and the generated texts remain the naturalness.

The proposed doesn't perform hyper-parameter search.

**Weaknesses:**

1. My main concern is the relationship of equations (4) and (5) and the claim on the last paragraph on page 4, "the positive action-masked policy considers only those actions for which the naturalness advantage is positive among the actions of the current policy, and the negative action-masked policy considers only those actions for which the naturalness advantage is negative among the actions of the current policy." I think the authors need to mathematically proof this claim is true.

2. Regarding to the experiments, I think the authors should provide the win rates of the proposed method vs several baselines by human evaluation. Current evaluations are based by human defined metrics like Rouge etc. which are problematic in real world applications.

**Questions:**

1. At the end of section 3.1: "the proportion of samples in the green area is inevitably very low" is there any proof or reference for this statement?

2. Why not include DPO as one of the baselines?

3. \gamma_natural in Equation (1) should be negative log ratio instead of KL divergence.

4. In section 2.1, the value and action-value functions should be formulated in a finite horizon instead of an infinite horizon for LLMs. Typo error, "The value and action-value function" where function should be "functions", plural

---

> ### Author Response · Authors · 2023-11-20
> **Response to Reviewer 99mP**
>
> Thank you for your constructive feedback and comments. Please feel free to ask any additional follow-up questions.
>
> **[Responses to Weaknessess]**
>
> 1. We apologize for any confusion caused by our explanation based on advantage function of naturalness (i.e. $A_{\text{natural}}^{\pi_\theta}$). The positive/negative action-masked policies defined in the paper do not consider actions whose naturalness advantage is positive/negative, but rather consider actions where $\pi_0 (s,a) - \pi_{\theta}(s,a)$ is positive/negative. To eliminate confusion, we removed the notations for $A_{\text{natural}}^{\pi_\theta}(s,a)$ and reorganized the equations more rigorously (**Section 2.2 and Section 3.1** in the revised paper). We also have added a more detailed explanation of how our main objective was derived based on the objective of PPO (**Appendix E** in the revised paper).
>
> 2. Thank you for suggesting the additional evaluation method. We will additionally conduct experiments and add it to the final version of the paper. However, the main purpose of our experiments is to show whether policy optimization with DfPO can successfully maximize task performance while maintaining naturalness. In the paper, we evaluated naturalness through various metrics such as perplexity, SPICE, and Bert score, so the current results are also meaningful to support our main claims.
>
> **[Responses to Questions]**
>
> 1. We experimentally observed that the proportion of samples in the green area is inevitably very low. This can also be seen through the tendency when learning the policy gradient (PG) methods including PPO. As can be seen from the learning curve of PPO or PG (Figure 1 or 5), task performance and perplexity increase as learning progresses, which means that most of the samples used for learning are in the area where $A_{\text{task}} > 0$ and $A_{\text{natural}} < 0$ (i.e. red area at the bottom left of Table 1).
>
> 2. DPO’s problem setting is different from the problem setting we consider in this paper, so it is difficult to compare fairly. (DPO is an offline RL algorithm that assumes a pairwise dataset is given, while ours is an online RL algorithm that assumes a learned reward function is given.)
>
> Although it is difficult to make a completely fair comparison with DPO, we additionally conducted experiments for DPO with settings as similar as possible. We present the experimental settings and results of DPO in the revised manuscript (**Appendix F**). Similar to the results of PPO, DPO also shows sensitive results depending on hyperparameter $\beta$ (**Figure 10** in **Appendix F.2**). Moreover, we also provide the perplexity-reward frontier for DPO, and the result shows that DfPO achieves better sentiment score performance than DPO for the same perplexity (**Figure 11** in **Appendix F.2**).
>
> 3. Thank you for pointing out the incorrect definition. We corrected it to a negative log ratio instead of KL divergence.
>
> 4. We revised the typos in the paper based on your comments.

---

> ### Author Response · Authors · 2023-11-22
> **Gentle Reminder**
>
> Dear Reviewer 99mP,
>
> Thanks again for your constructive feedback and comments. As the discussion period is soon to end, we would like to know if your questions and concerns have been resolved by our response and revision. We would be grateful if you could offer feedback on our rebuttal, and if your questions and concerns have been resolved by our rebuttal, we hope you update your rating accordingly. We appreciate the reviewer's insightful and constructive comments once again.
>
> Best,
>
> Authors

---

### Official Review · Reviewer_hR6e · 2023-11-01

**Soundness:** 3 good
**Presentation:** 3 good
**Contribution:** 3 good
**Rating:** 8
**Confidence:** 3

**Summary:**

The paper proposes a new reinforcement learning method called Degeneration-free Policy Optimization (DfPO) for fine-tuning language models without causing text degeneration.

It observes that standard RL methods like PPO are prone to text degeneration when fine-tuning LMs, due to unbalanced optimization of task rewards vs language model likelihood.

DfPO introduces an "action-masked policy" to avoid sampling tokens that can cause unexpected optimization.

It uses separate clipped advantage functions to perform stable likelihood maximization and minimization.

Experiments on text generation benchmarks show DfPO improves task scores while preserving fluency, without needing sensitive KL penalty hyperparameters like PPO/NLPO.

DfPO outperforms PPO/NLPO on generation quality even without hyperparameter tuning.

In summary, the main contributions are proposing DfPO to address text degeneration in LM fine-tuning, introducing techniques like action masking and clipped advantages, and demonstrating improved performance over standard RL methods empirically.

**Strengths:**

Here is an assessment of the key strengths of this paper:

1. The paper addresses an important problem in language model fine-tuning - text degeneration when using reinforcement learning. This is a major challenge limiting the applicability of RL for optimizing language models.

2. The proposed method DfPO introduces some novel and creative ideas to tackle this problem:

     a. Using action masking to avoid sampling risky tokens

     b. Separate clipped advantage functions for stable likelihood optimization

     c. Policy optimization via likelihood maximization/minimization

3. These techniques seem technically sound and are motivated clearly. The overall algorithm is also clearly explained through figures, equations and pseudocode.

4. The experimental methodology is very thorough, rigorously comparing DfPO against strong baselines like PPO, NLPO on standard text generation benchmarks.

5. Results demonstrate clear improvements in balancing task rewards versus fluency, without needing extra hyperparameter tuning.
The paper is well-written, laying out the background, approach, experiments in a structured and easy to follow manner.

6. Solving text degeneration could significantly expand the applicability of RL for language tasks. So the work has very good potential for real-world impact.

In summary, the paper makes solid technical contributions through creative ideas, and supports them through extensive experiments and clear writing. Given the significance of the problem it addresses, this work could be impactful for the field.

**Weaknesses:**

Overall, the paper makes good technical and experimental contributions. But expanding the theoretical analysis, testing on more tasks and models, and providing more discussion would further strengthen the paper. The weaknesses are more about opportunities for extending the work rather than flaws in what has been done.

1. While the core DfPO techniques seem sound, more analysis could be provided on the dynamics of how action masking and separate advantages control optimization.

2. Only text generation tasks are evaluated. Applying DfPO to other language tasks like summarization could strengthen the generality of the approach.

3. More analysis could be provided on the diversity of generated text - perhaps measuring repetitiveness.

**Questions:**

None

---

> ### Author Response · Authors · 2023-11-20
> **Response to Reviewer hR6e**
>
> Thank you for your constructive feedback and comments. Please feel free to ask any additional follow-up questions.
>
> **[Responses to Weaknessess]**
>
> 1. Thank you for the comments on more theoretical and experimental analysis. To explain our method based on a more rigorous mathematical formulation, we added a more detailed explanation of how our main objective was derived based on the objective of PPO (**Appendix E** in the revised paper). Moreover, to analyze our method, we also provide an ablation study in **Appendix C.1**.
>
> 2. Thank you for suggesting the experiments for other language tasks. We will additionally provide results for the other tasks in the final version of the paper.
>
> 3. Thank you for suggesting more analysis on the diversity of generated text. Our paper already provides the experimental results on the diversity of generated text (Tables 3 and 5). We will additionally add more analysis on it including measuring repetitiveness.

---

> ### Author Response · Authors · 2023-11-22
> **Gentle Reminder**
>
> Dear Reviewer hR6e,
>
> Thanks again for your constructive feedback and comments. As the discussion period is soon to end, we would like to know if your questions and concerns have been resolved by our response and revision. We would be grateful if you could offer feedback on our rebuttal, and if your questions and concerns have been resolved by our rebuttal, we hope you update your rating accordingly. We appreciate the reviewer's insightful and constructive comments once again.
>
> Best,
>
> Authors

---

### Official Review · Reviewer_wZxt · 2023-11-09

**Soundness:** 1 poor
**Presentation:** 1 poor
**Contribution:** 2 fair
**Rating:** 3
**Confidence:** 3

**Summary:**

For language generation, naively using RL to optimize task score may lead to unnatural outputs from the learned language model. The standard practice to deal with this problem is to regularize the RL training with a weighted sum of the task reward and a KL-penalty, which however introduces additional hyperparameters (the weight) that need to be carefully tuned. This paper proposes an alternative method, called DfPO in the paper, which does not use the KL weight to reconcile the optimization trade-off here -- thus avoiding additional hyperparameter -- but instead will only update the policy gradient on state-actions where the task reward and the KL-penalty can be consistently optimized. The paper shows that their algorithm can achieve similar performance with PPO and NLPO, but without hyperparameter tuning, on two tasks of the GRUE benchmark.

**Strengths:**

The current tricks and hyperparameters involved to constrain RL-trained language models from deviating "too far" from the pre-trained model is, in my opinion, an "ugly" part of the practice. So, I appreciate much of this paper's aspiration to try solving the problem without additional hyperparameters, and hopefully, solving it in a more elegant way.

It appears that the proposed algorithm has almost constant perplexity on one task, even though the algorithm itself does not seem to explicitly impose this. It would be an interesting property if the same observation applies in more general scenarios.

**Weaknesses:**

**(a)** I have a number of concerns and confusions about the sections that narrate the proposed algorithm and its rationale. Many important statements are claimed but not properly justified or proved. Math equations are sometimes sloppy. The pseudo-code of the proposed algorithm also left important details unclear to me. See my Question 1 ~ 10 below for the detailed concerns. Overall, I am not sure that the algorithm and its explanation in the paper indeed make sense.

**(b)** I also have concerns about the experiment part. In the abstract, the introduction section, the conclusion section, as well as in the experiment section (Section 5.1, the paragraph starting with "Comparison with baselines", page 8), the paper repeatedly and explicitly states that the proposed algorithm DfPO "outperforms PPO and NLPO". I think Table 2 is the main supportive evidence for this claim. However, the sentiment score improvement there (+0.01) is within 1-stdev (and the stdev is from 5 runs only) so the improvement may not be statistically significant. Regarding perplexity score, although DfPO is indeed 1.3 lower, I notice that in Table 2 the supervised learning method actually leads to the largest increase of perplexity, and as supervised learning is not known to hurt naturalness significantly, I wonder if the small difference in perplexity score here can indeed translate to substantial improvement on naturalness or not. Overall, I'm not sure that the experiment results well support the main claim that DfPO outperforms.

Besides that, some numerical results in the experiment part are also subject to questions. See my Question 11 ~ 14 below for details.

Finally, the proposed algorithm was only tested on two tasks of the GRUE benchmark. Given my major concerns on the rationale of the algorithm, I expect more comprehensive empirical evaluations and from the current experiment I am not confident that the algorithm will perform well in more general scenarios.

**Questions:**

1. My impression, after reading Section 2 and 3, is that you define the "degeneration problem" as the reduced "naturalness" and that you measure naturalness with the KL-divergence against the initial policy $\pi_0$ which is a pre-trained language model. However, being different from the particular initial model does not mean that the new model's output is unnatural. In fact, any effective improvement over the initial model has to output differently from it, thus *necessarily* increasing the KL-divergence, right? In general, although we seek for naturalness through controlling the KL-divergence, we don't really aim at *minimizing* it, which is why we have it weighted by $\beta$ in standard practice. So, although it works as a heuristic trick, I doubt if the KL-divergence can *represent* the naturalness.

2. I feel, in many places of the paper, you are assuming that $A_{natural}^{\pi_\theta}(s,a)>0$ is equivalent to $\pi_0(s,a) > \pi_\theta(s,a)$. If this is the case, can you elaborate this equivalence more, or better, prove it formally? If this is not the case, then in the last paragraph of Page 4, why did you say "In other words, the positive action-masked policy considers only those actions for which the naturalness advantage is positive among the actions of the current policy", while in Eq.(4) the positive action-masked policy is defined to only consider actions with $\pi_0 > \pi_\theta$? This is just an example, there are many other places where the paper is hinting about the equivalence, to my current understanding.

3. Page 4, in the last paragraph of Section 3.1, you said "state-actions in the red area ... are undesired samples that can cause degeneration". Any evidence that these samples have indeed *caused* the degeneration problem? As a guess, I feel you might be thinking that: actions with $A_{natural} > 0$ will have $\pi_0 >\pi_\theta$, and so, for the purpose of minimizing the KL-divergence we should increase the probability of this action, which however hurts the task score as $A_{task}<0$ on this action. If this is the argument, I'm not sure that actions with $A_{natural} > 0$ indeed have (or even tend to have) $\pi_0 >\pi_\theta$, which is exactly what I'm asking to prove in last question. Note that an action does not have fixed KL-penalty; when the policy changes, the KL-penalty on the same action changes too. This is in sharp difference to the task reward, which is constant for a given action regardless of the policy.

4. Eq.(6), I am not sure about the equation here, can you prove it? It seems to be the central theoretical result of the paper but I can't find any justification of the equality here.

5. Page 5, in the last paragraph of Section 3.3, you said your algorithm eventually outputs the negative policy after the training because it "consists of actions with high task performance while preserving naturalness". However, actions generated by the negative policy are actually those that we want to *minimize* their likelihood in the policy for the sake of "high task score and naturalness", why do we want to output them now?

6. In the pseudo-code of Algorithm 1, what do you mean "running policy" at line 4? Do you mean rolling out the policy in the environment with initial state $s_0^p$? But in that case the given training data D will only be used to provide initial states, with all the subsequent data in the actual training separately sampled from the environment, is this correct understanding? Or do you mean re-sample a trajectory over the given dataset D using the positive or negative policy (but how)?

7. Again in the pseudo-code of Algorithm 1, you are using the data from both the positive and the negative policies to train V, so V is the value function of what policy? Also, from the pseudo-code I can't find how V is used at all. I assume V will be used to estimate the advantage functions, but there are two advantage functions, $A^{\pi^+}$ and $A^{\pi^-}$, which are the advantage with respect to different policies, how do we use the single value function (of a unknown policy) to estimate two different advantage functions?

8. In the last paragraph of Section 3.1 you said "the proportion of samples in the green area is inevitably very low, which is inefficient in terms of sample efficiency for learning". However, by clipping the advantage at 0 isn't your algorithm essentially only using samples in the green area? Specifically, in the first term of Eq.(6), for an action $a \sim \pi^+$, it must have $\pi_0(a)>\pi_\theta(a)$; when such an $a$ has $A_{task}(a)<0$, its clipped advantage is 0 thus its contribution to the gradient is 0. This is equivalent to throwing away the actions in the corresponding red area, which is inefficient as you pointed out in the paper. In fact, in the same paragraph you also said "the first term in Eq.(6) corresponds to the likelihood maximization [entry] in Table 1", which seems to echo my reasoning above. Did you compare the sample efficiency of your algorithm with PPO and NLPO in the experiment?

9. In Eq.(1), what is the $KL$ function here? It cannot be the usual KL-divergence function as KL-divergence applies to two distributions while in Eq.(1) the $KL$ function is applied to two probability values ($\pi_\theta(a_t|s_t)$ and $\pi_0(a_t,s_t)$).

10. In Eq.(2), $s$ is an open variable that is not averaged out, thus the equation is not really correct. Also, the objective function $J$ is not explicitly defined before it shows up in this equation. The paragraph right above this equation gives $E[\sum_t r(s_t,a_t)]$ as the objective function, where the discount factor $\gamma$ is dropped, in which case the policy gradient formula would be a bit more trickier than Eq.(2). Please double check.

11. In Table 2, how did you choose a single "best" PPO/NLPO result given that there are two metrics? In Figure1 the PPO's sentiment score can be as high as 0.85. Although that particular PPO instance is at the cost of high perplexity, it does let me wonder whether there exists *other* hyperparameter of PPO that can achieve some sentiment between 0.6 and 0.8 and yet have reasonable perplexity.

12. Any explanation why the 6B-model's performance (0.60- in Figure 6) is lower than the 100M-model's performance (0.62+ in Table 2) when trained with DfPO?

13. Table 6, what does the "batch size" correspond to in Algorithm 1? N or M or something else?

14. The perplexity curve on the IMDB task appears to be very flat. Is this also the case in other task? Can you show the curve for Commonsense Generation, for example? I guess the perplexity score here is over the GPT-2 outputs, did you also examine the perplexity over the reference outputs in the testing data (which should be also very natural human language), is the curve also as flat?

---

> ### Author Response · Authors · 2023-11-20
> **Response to Reviewer wZxt (1/2)**
>
> Thank you for your constructive feedback and comments. Please feel free to ask any additional follow-up questions.
>
> **[Responses to Weaknessess]**
>
> (a) Thank you for pointing out some equations and statements that are not rigorously defined and justified. We revised the paper to explain more clearly the process by which our main objective was derived, and revised the equations to be more rigorously (**Section 2.2, 3.1 and Appendix E** in the revised paper).
>
> (b)  Thank you for clarifying the contribution of our work. As you pointed out, our main goal is not to outperform PPO/NLPO obtained through hyperparameter tuning. The main contribution of our paper is to **obtain similar performance (without any hyperparameter search**) to PPO/NLPO obtained by hyperparameter search. We revised our manuscript to clarify the contribution of our paper accordingly (from "outperforms PPO/NLPO" to "achieves similar performance to PPO/NLPO without any hyperparameter search").
>
> **[Responses to Questions]**
>
> 1. As you mentioned, being *different* from the particular initial model does *not* mean that the new model’s output is *unnatural*, but being *close* from the particular initial model means that the new model’s output is *natural*. We agree that KL divergence cannot perfectly represent naturalness, and we don’t aim at *minimizing* KL divergence. We aim to improve the task performance while preventing too much deviation from the initial model based on the log ratio with the initial model. Our goal is similar to existing methods that use log ratio as reward penalty, but **we constrain action support with a masked policy instead of reward penalty**.
>
> 2. We apologize for the confusion caused by our explanation based on the advantage function of naturalness (i.e. $A_{\text{natural}}^{\pi_\theta}(s,a)$). It is correct that our purpose was to imply $\pi_0(s,a) > \pi_{\theta}(s,a)$ (NOT $A_{\text{natural}}^{\pi_\theta}(s,a)$ > 0), and according to the strict definition of the advantage function, these are not equivalent. Therefore, to eliminate the confusion, we removed the notations for $A_{\text{natural}}^{\pi_\theta}(s,a)$ and reorganized the equations more rigorously (**Section 2.2 and Section 3.1** in the revised paper). We also have added a more detailed explanation of how our main objective was derived based on the objective of PPO (**Appendix E** in the revised paper).
>
> 3. As in the previous answer, we removed the notations for $A_{\text{natural}}^{\pi_\theta}(s,a)$ and reorganized the equations more rigorously. Please see the previous answer and **Section 2.2, 3.1 and Appendix E** in the revised paper.
>
> 4. Our DfPO objective is derived from a policy gradient method with a heuristic modification, which is closely related to the heuristic used in PPO. We added a more detailed explanation of how our main objective was derived based on the objective of PPO (**Section 2.2 and Appendix E** in the revised paper).
>
> 5. To summarize what happens when likelihood maxi/minimization is performed with (s,a) sampled from positive/negative action-masked policy, it is as follows:
>
> (1) Likelihood maximization with samples from positive action-masked policy:
>
> - Increases the likelihood of the $(s,a)$ pair, which will improve the task performance of the policy (i.e. $A_{\text{task}}^{\pi_{\text{mask}}^+} (s,a) > 0$).
>
> - It means that $(s,a)$ pairs that will improve the task performance of the policy gradually become $\pi_{\theta}(s,a) > \pi_0(s,a)$.
>
> (2) Likelihood minimization with samples from negative action-masked policy:
>
> - Decreases the likelihood of the $(s,a)$ pair, which will deteriorate the task performance of the policy (i.e. $A_{\text{task}}^{\pi_{\text{mask}}^-} (s,a) < 0$).
>
> - It means that $(s,a)$ pairs that will deteriorate the task performance of the policy gradually become $\pi_{\theta}(s,a) < \pi_0(s,a)$.
>
> The above likelihood max/minimization can perform policy optimization while maintaining naturalness, and good actions for the task performance gradually satisfy $\pi_{\theta}(s,a) > \pi_0(s,a)$, which is equivalent to sample from negative action-masked policy. This is why we use the negative action-masked policy for the inference, and we also have experimentally observed that the task performance deteriorates when inference is performed with other policies ($\pi_\theta$ and positive action-masked policy) rather than the negative action-masked policy. We added this result to **Appendix C.6** in the revised paper.
>
> 6. Yes, you are right. The "running policy" means rolling out the policy in the environment with the initial state $s_{0}^{p}$. Similar to when training PPO/NLPO in [1], the given training data is only used to provide initial states.
>
> [1] Rajkumar Ramamurthy and Prithviraj Ammanabrolu, et al. "Is Reinforcement Learning (Not) for Natural Language Processing: Benchmarks, Baselines, and Building Blocks for Natural Language Policy Optimization." ICLR 2023.

---

> ### Author Response · Authors · 2023-11-20
> **Response to Reviewer wZxt (2/2)**
>
> 7. Thank you for pointing out the typo. V in algorithm 1 is a typo. The value functions are also separately defined for $\pi_\text{mask}^+$ and $\pi_\text{mask}^-$ (i.e. $V_{\phi}^{\pi_\text{mask}^+}$ and $V_{\phi}^{\pi_\text{mask}^-}$), and is used for advantage function estimation corresponding to $A_{\text{task}}^{\pi_\text{mask}^+}$ and $A_{\text{task}}^{\pi_\text{mask}^-}$. We revised the typos in the paper based on your comments.
>
> 8. We experimentally observed that the proportion of samples in the green area is inevitably very low. This can also be seen through the tendency when learning the policy gradient (PG) methods including PPO. As can be seen from the learning curve of PPO or PG (Figure 1 or 5), task performance and perplexity increase as learning progresses, which means that most of the samples used for learning are from the area where $A_{\text{task}}^{\pi_\theta} > 0$ and $\pi_0 (s,a) < \pi_{\theta}(s,a)$ (i.e. red area at the bottom left of Table 1). Therefore, when sampling from $\pi_{\theta}$, the proportion of samples in the green area is very low. This is why we use Equation (13) as the final objective, rather than Equation (12). We also added this explanation to **Appendix E.2** in the revised paper.
>
> 9. Similar to other papers ([1], [2]), we used negative log ratio instead of KL divergence. We corrected the paper from KL divergence to a negative log ratio.
>
> 10. Thank you for pointing out some equations that are not rigorously defined. Based on your comments, we revised the paper to express the equations more rigorously.
>
> 11. All PPO/NLPO results are from NLPO paper [1], and are the best results selected by the authors considering the trade-off between sentiment score and perplexity. (We evaluated and recorded DfPO results with exactly the same settings as the PPO/NLPO experiments.) As mentioned in your question, it is very difficult to choose the "best" results of PPO/NLPO given that there are two metrics, and alleviating this **ambiguity in model selection** is one of the main contributions of our work.
>
> 12. Since the reference policies are different (as each initial model), a larger model does not guarantee that task performance will always improve faster. In other words, in the case of the 6B-model, policy optimization is performed while maintaining naturalness at the level of initial 6B-model, so the speed of improvement in task performance may be relatively slow. However, when compared with the same iteration (iteration: 100), 100M and 6B models have similar task performance (0.60-), and 6B-model is not slower.
>
> 13. The “batch size” in Table 6 corresponds to M in Algorithm 1.
>
> 14. The perplexity scores recorded in the paper are evaluated as the perplexity over the reference output in the testing data (NOT over the GPT-2 outputs). Additionally, we added the learning curves of commonsense generation task for both task score (i.e. Meteor) and naturalness (i.e. Perplexity) to **Appendix C.4** in the revised paper. Similar to the results of IMDB task, task score (Meteor) is improved while naturalness (Perplexity) is maintained. Moreover, in the Commonsense Generation task, we also evaluated naturalness through the BertScore and Spice, and the results in Table 4 show that task performance is improved while maintaining naturalness.
>
> [1] Rajkumar Ramamurthy and Prithviraj Ammanabrolu, et al. "Is Reinforcement Learning (Not) for Natural Language Processing: Benchmarks, Baselines, and Building Blocks for Natural Language Policy Optimization." ICLR 2023.
> [2] Rafael Rafailov, et al. “Direct Preference Optimization: Your Language Model is Secretly a Reward Model.” NeurIPS 2023.

---

> ### Author Response · Authors · 2023-11-22
> **Gentle Reminder**
>
> Dear Reviewer wZxt,
>
> Thanks again for your constructive feedback and comments. As the discussion period is soon to end, we would like to know if your questions and concerns have been resolved by our response and revision. We would be grateful if you could offer feedback on our rebuttal, and if your questions and concerns have been resolved by our rebuttal, we hope you update your rating accordingly. We appreciate the reviewer's insightful and constructive comments once again.
>
> Best,
>
> Authors

---

### Author Response · Authors · 2023-11-20
**General Response**

We thank all the reviewers for their constructive feedback and comments. Below we summarize the improvements and modifications we made through the rebuttal. If you have any additional questions or concerns to our response, we are happy to provide additional responses during the rebuttal period.

The detailed improvements and modifications in the revision are summarized as follows:

**[Experiments]**

- Additional results and analysis for the additional baselines (DPO and PPO+SFT) to address the suggestions provided by Reviewer **99mP** and **6zCc** (DPO in **Appendix F** and PPO+SFT in **Appendix C.5**).
- Additional results with varying policy for inference to address the questions provided by Reviewer **wZxt** and **6zCc** (**Appendix C.6**).
- Additional results of learning curve on commonsense generation task to address the questions provided by Reviewer **wZxt** (**Appendix C.4**).

**[Writing]**

- Clarification of our contribution (change "outperforms PPO/NLPO" to "achieves similar performance to PPO/NLPO without any hyperparameter search") to address the suggestions provided by Reviewer **wZxt** (**Abstract, Section 1, 5, and 6**).
- Reorganization for more rigorous explanation to address the suggestions provided by Reviewer **wZxt** (**Section 2, 3**).
- Detailed explanation of how our main objective was derived based on the objective of PPO to address the questions provided by Reviewer **wZxt** (**Appendix E**).
- To eliminate the confusion, we removed the notations for $A_{\text{natural}}^{\pi_\theta}(s,a)$ and reorganized the equations more rigorously to address the questions provided by Reviewer **wZxt** and **99mP** (**Section 2, 3**).

---

### Meta-Review · Area_Chair_rHED · 2023-12-05

**Metareview:**

Previous RL typically optimizes a task-specific reward and a KL divergence, which requires tuning a sensitive hyperparameter balancing the two terms. This paper conducted some analysis and proposed a constrastive learning-like objective to eliminate the need of hyperparameter tuning.  Although the approach does not achieve astonishing performance improvement, reviewers generally recognize that the approach is neat and valuable.

However, the paper appears sloppy in many aspects, including mathematical formulations. In the revision, the authors made significant changes, but me and reviewers don't have too much confidence at this moment. I would encourage the authors to consider the suggestions and incorporate them in the next version.

**Justification For Why Not Higher Score:**

The paper appears sloppy in many aspects, including mathematical formulations. In the revision, the authors made significant changes, but me and reviewers don't have too much confidence at this moment.

**Justification For Why Not Lower Score:**

N/A

---

### Decision · Program_Chairs · 2024-01-16

Reject